



# Updated climatological mean delta fCO$_2$ and net sea–air CO$_2$ flux over the global open ocean regions

Amanda R. Fay[1], David R. Munro[2,3], Galen A. McKinley[1], Denis Pierrot[4], Stewart C. Sutherland[1], Colm Sweeney[3], Rik Wanninkhof[4]

Affiliations:

[1] Columbia University and Lamont-Doherty Earth Observatory, Palisades, NY, USA
[2] Cooperative Institute for Research in Environmental Sciences (CIRES), University of Colorado, Boulder, CO, USA
[3] Global Monitoring Laboratory, National Oceanic and Atmospheric Administration, Boulder, CO, USA
[4] Atlantic Oceanographic and Meteorological Laboratory, National Oceanic and Atmospheric Administration, 4301 Rickenbacker Causeway, Miami, FL, USA

Corresponding author: Amanda R. Fay afay@ldeo.columbia.edu

Key Points
- An updated surface water CO$_2$ climatology for 1980-2021 is created using the SOCAT database, following procedures of Takahashi et al. (2009)
- A net air-sea CO$_2$ flux of -1.79 PgC yr-1 is determined for near-global ocean coverage



# Abstract

The late Taro Takahashi (LDEO/Columbia University) provided the first near-global monthly air-sea $CO_2$ flux climatology in Takahashi et al. (1997), based on available surface water partial pressure of $CO_2$ measurements. This product has been a benchmark for uptake of $CO_2$ in the ocean. Several versions have been provided since, with improvements in procedures and large increases in observations, culminating in the authoritative assessment in Takahashi et al. (2009). Here we provide and document the last iteration using a greatly increased dataset (SOCATv2022) and determining fluxes using air-sea partial pressure differences as a climatological reference for the period 1980-2021 (Fay et al. 2023). The resulting net flux for the open ocean region is estimated as -1.79 PgC yr$^{-1}$ which compares well with other global mean flux estimates. While global flux results are consistent, differences in regional means and seasonal amplitudes are discussed. Consistent with other studies, we find the largest differences in the data-sparse southeast Pacific and Southern Ocean.

# 1. Introduction

Atmospheric carbon dioxide ($CO_2$) levels now exceed 415 ppm on an annual basis and the continued growth of the atmospheric reservoir represents a major societal concern due to the impact on the radiative balance of the atmosphere. Warming and associated environmental changes including sea-level rise and ocean acidification have adverse effects on countless aspects of terrestrial and marine ecosystems which in turn impact air-sea exchange of $CO_2$ and the trajectory of the atmospheric $CO_2$ levels. The annually updated Global Carbon Budget (GCB) report estimates current net global ocean carbon uptake has been estimated at nearly 3.0 PgC per year, which corresponds to about a quarter of the total annual emissions (the total anthropogenic $CO_2$ emission, including the cement carbonation sink, is estimated at 10.9 ± 0.8 PgC yr$^{-1}$) (Friedlingstein et al. 2022). Given the importance of the ocean as a $CO_2$ sink, it is essential to continuously monitor changes and improve our understanding of the ocean's role in the global carbon cycle.

Over the last several decades, multiple approaches have been developed to measure the impact of the ocean on the global $CO_2$ cycle. These approaches include atmospheric inversions (Feng et al. 2019), global atmospheric $O_2/N_2$ (Manning & Keeling 2006), $^{13}C$ measurements (Quay et al. 1992, Tans et al. 1993), ocean inventory approaches (Gruber et al. 2023) and the measurement of surface ocean and atmospheric $CO_2$ (Takahashi et al. 1993); all methods work towards the goal of elucidating the net flux of $CO_2$ from the atmosphere into the ocean. These different



approaches have multiple advantages and disadvantages depending on the time and spatial scale of interest. Directly measuring surface ocean and atmospheric $CO_2$ levels has the advantage, given sufficient measurements, of deriving spatial and temporal variability over the ocean surface on short temporal and spatial scales. This method provides valuable insights into key processes driving the uptake and emissions of carbon when combined with our understanding of ocean physics and biological activity.

The surface ocean and atmospheric $CO_2$ approach leverages available observations and the dynamic sea-air gradient between the partial pressure of carbon dioxide ($pCO_2$) in the surface ocean ($pCO_2^{oce}$) and the overlying atmosphere ($pCO_2^{atm}$), known as the delta $pCO_2$ ($\Delta pCO_2$) and typically defined as $pCO_2^{oce}$ - $pCO_2^{atm}$. This difference is the thermodynamic driving force for the transfer of $CO_2$ into (negative) and out of (positive) the ocean. On average, the $\Delta pCO_2$ across the global oceans are becoming increasingly negative as atmospheric $CO_2$ levels steadily rise, leading to an increasing carbon sink. Limited regions around the globe are sources of $CO_2$ to the atmosphere, including the equatorial Pacific Ocean and other areas of persistent upwelling.

The late Taro Takahashi was a leader in efforts to characterize air-sea $CO_2$ flux through the design and deployment of $pCO_2$ systems throughout the global oceans, and perhaps most importantly, his efforts to assemble, evaluate and construct global ocean climatologies from available $pCO_2^{oce}$ datasets. These versions of the ocean $pCO_2$ climatology have been presented in literature (Takahashi et al. 1997, 2002, 2009, and 2014), henceforth referred to as T-1997, T-2002, T-2009, and T-2014; all of which have been highly utilized and cited by carbon cycle researchers from around the world. We present an updated near-global climatological mean distribution and net sea–air $CO_2$ flux which represent the mean of ocean conditions over the last four decades. This climatology is unique compared to other advanced machine learning approaches (e.g., Rödenbeck et al. 2015) in that it interpolates in time and space using only the available $pCO_2$ data rather than using proxy variables for gap filling. This difference in methodology provides a valuable alternative approach to the ongoing effort to characterize the global ocean carbon sink. This benchmark is critical for global carbon assessments, notably the Regional Carbon Cycle Assessment and Processes (RECCAP2, DeVries et al. 2023) effort.

Building on previous work of Takahashi and colleagues, we employ the same time-space interpolation method used in the previous versions of the Takahashi climatology (e.g., T-2002, T-2009, T-2014) to create the climatology. However, here we use the Surface Ocean $CO_2$ Atlas (SOCAT) v2022 database (Bakker et al. 2016, 2022), rather than the LDEO database curated by Taro Takahashi. We use the SOCAT database for this update because it is the most comprehensive database of available observations



from international research groups. We have included the climatology produced using the most recent LDEO database (LDEOv2019, Takahashi et al. 2020) with data extending to 2019, in supplementary figures and text but our main findings will focus on
the results from the SOCAT database.

# 2. Data

## 2.1 SOCAT database

The SOCAT database was first released in 2013 (Pfeil et al. 2013) and is updated
annually (Bakker et al. 2016). Observations are reported as values of fugacity of $CO_2$ ($fCO_2$) in micro atmospheres (µatm), along with a collection of ancillary data including concurrent observations such as SST (more accurately, the ship intake temperature), temperature of equilibration, salinity, and sea level pressure at the time of equilibration. The SOCAT database also includes supplemental variables with values interpolated
from gridded global datasets such as satellite SST and NCEP sea level pressure. The database restricts the included data to only observations that are measured in near-continuous operation or in discrete samples with an equilibrator system; specifically, it does not include $fCO_2$ values that are calculated from other ocean carbon measurements including dissolved inorganic carbon, total alkalinity and/or pH. More
information on the SOCAT database is available in Bakker et al. (2016); current and previous releases are available to download at https://www.socat.info/index.php/data-access/.

The SOCAT database (Bakker et al. 2016) is the largest and most widely used
collection of quality-controlled $fCO_2$ data with over twice the number of observations included in the latest LDEO database (LDEOv2019). In this work, we utilize the SOCATv2022 release which includes over 33.7 million observations spanning the years 1957 through 2021 (DOI:10.25921/1h9f-nb73, accessed on July 15, 2022, Bakker et al., 2022). We use observations beginning in 1980 due to limited metadata available for
earlier observations. This time restriction eliminates only 24,786 observations, or less than 0.1% of the total number of observations included in the SOCATv2022 release. We also exclude coastal observations collected within 100km of land similar to past LDEO climatologies which reduces the total number observations utilized to just over 21.3 million. Unlike past LDEO climatologies, this climatology does not exclude observations
collected in the equatorial Pacific during El Niño periods.

## 2.2 fCO$_2$ vs pCO$_2$

For this climatology, we report values of the fugacity of carbon dioxide (fCO$_2$) rather than pCO$_2$. The fCO$_2$ is equal to the pCO$_2$ corrected for non-ideality of CO$_2$ solubility in water using the virial equation of state (Weiss 1974). The fugacity correction for surface water is 0.996 and 0.997 at 0 ºC and 30 ºC respectively (Dickson et al. 2007), or 0.7 to 1.2 µatm lower than the corresponding pCO$_2$, and depends primarily on temperature for the conversion although pressure is also included in the conversion equation. It is now common practice in the observational community to present observed values as fCO$_2$ and this option has been endorsed by the IOCCP (International Ocean Carbon Coordination Project). The correction of pCO$_2^{oce}$ and corresponding pCO$_2^{atm}$ values to fCO$_2$ is practically identical such that the resulting $\Delta$fCO$_2$ is always within 0.1 µatm compared to the corresponding $\Delta$pCO$_2$. As a result, this difference will not have a meaningful impact on air-sea flux calculations. Only at elevated fCO$_2$ levels, such as those in the subsurface ocean, is the difference between fCO$_2$ and pCO$_2$ significant. Therefore, the shift in this climatology from pCO$_2$ to fCO$_2$ simply aligns this updated climatology with current community best practices. This choice avoids conversions given that the SOCAT database reports fCO$_2$ values.

## 2.3 Distribution of measurements

At present, the SOCAT database relies on voluntary submission of quality controlled data from over 100 scientists. The number of observations has increased significantly over past decades, facilitated by a now-automated data submission process. Even with this increase in observations, there are only a few regions over the global oceans where fCO$_2^{oce}$ has been systematically monitored over multiple decades at nearly the same location (Bakker et al. 2016; Bates et al. 2014; Landschützer et al. 2016, Figure 1). Of the observations considered in this analysis, spanning years 1980-2021, only 1.4% of the monthly 1º by 1º global ocean grid cells have measured values. Most of the data (65%) was collected since 2010.

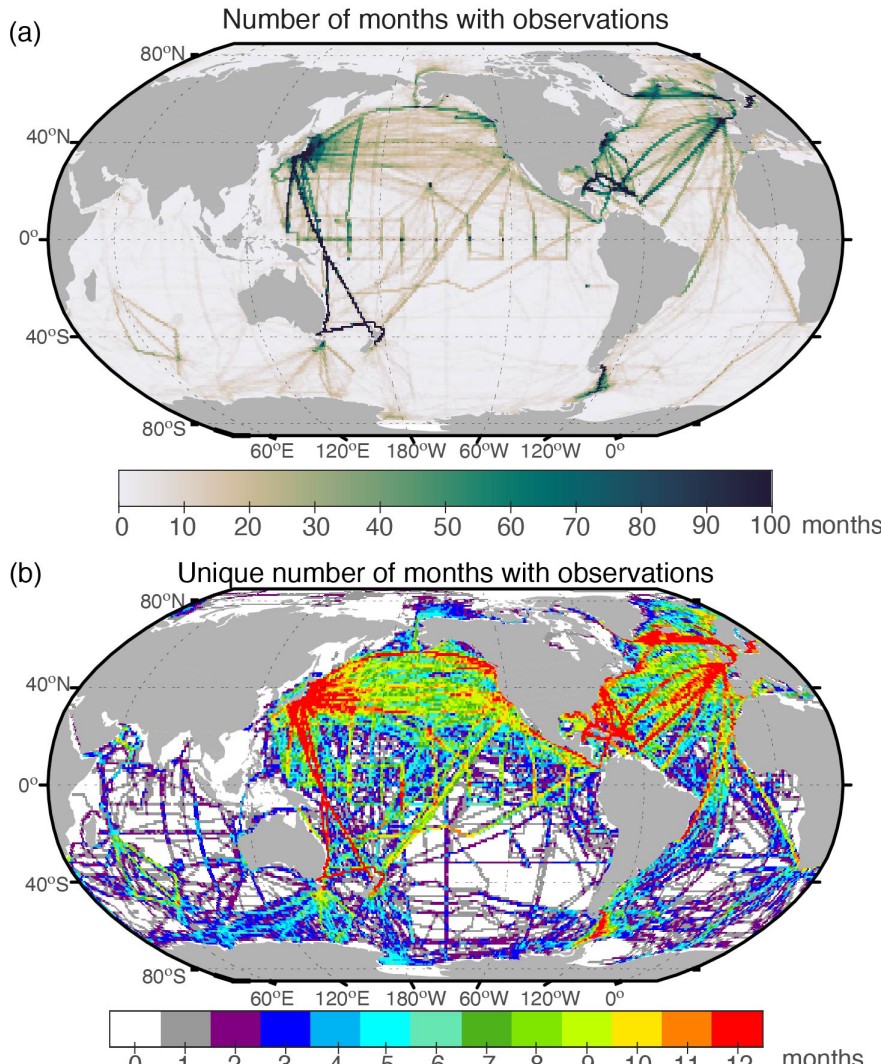

Figure 1: (a) Total number of months with at least one observation in each 1° grid cell in the SOCATv2022 database, for years 1980-2021 (Bakker et al. 2022). The maximum number possible for a grid cell is 504 (42 years * 12 months). (b) the number of unique calendar months in each grid cell where at least one observation has been made since
1980. Red indicates grid cells where each month (Jan - Dec) has been sampled at least once over the 40+ year time series while white indicates grid cells with no measurements over the length of the time series.




An additional challenge to global monitoring efforts is that observations are not collected
consistently throughout the annual cycle in many locations around the globe, thus
requiring considerable interpolation to produce a full seasonal climatology. Much of the
ocean contains data collected in fewer than three unique months of the year, regardless
of how many years of data is available (Figure 1b). Current efforts utilize proxy variables
and machine learning to identify relationships between ocean carbon and better-
observed variables (often SST, chlorophyll, mixed layer depth, etc.) and then through
those relationships extrapolate available ocean $fCO_2$ to fill the missing months of the
seasonal cycle. Unlike those methods (e.g., methods compared by Rödenbeck et al.
2015), we do not utilize any proxy variables in this method and only rely on available
$fCO_2^{oce}$ values in the SOCAT database to estimate the seasonal climatology. Hence,
this effort provides a complementary alternative interpolation method to other
approaches.

## 2.4 Atmospheric $fCO_2$

For the calculation of atmospheric carbon dioxide, we utilize zonally invariant NOAA
marine boundary layer (MBL) $xCO_2$ values which are reported in units of ppm or
µmol/mol (Lan et al. 2023) and provided with each observation in the SOCAT dataset.
In order to calculate $fCO_2^{atm}$ values from MBL $xCO_2^{atm}$ values, we follow standard
operating procedures and equations outlined in Dickson et al. (2007) and use the SST,
salinity and sea level pressure observations also reported for each value in the SOCAT
database (Bakker et al. 2022). The SST (more specifically, ship intake temperature) is
measured concurrently with surface ocean $CO_2$, sea level pressure is from the National
Centers for Environmental Prediction (NCEP), and surface salinity is from the World
Ocean Atlas (WOA). Delta $fCO_2$ ($\Delta fCO_2$) is calculated by subtracting corresponding
atmospheric from ocean values ($fCO_2^{oce}$ - $fCO_2^{atm}$).

# 3. Methods

## 3.1 Normalization to a reference year

In previous versions of the LDEO climatology, emphasis was placed on the calculation
of trends in surface ocean carbon levels for all regions of the global ocean. These
trends were used to normalize all available observations to one reference year by
correcting for the estimated change that would be expected between the collection date
of the observation and the reference year. In this updated climatology, we use $\Delta fCO_2$



values as input to the algorithm rather than allowing for the adjustment of $fCO_2^{oce}$ to a specific reference year. A similar methodology was used in early versions of the LDEO climatology (i.e., T-1997).


By utilizing this method to collapse all available data to one year, we make the assumption (as made by T-1997) that the ocean and atmosphere are changing at the same rate and thus the $\Delta fCO_2$ has been constant over the 40+ years of observations. This assumption allows for a standard method for the normalization of all observations
to one calendar year. This method is utilized in contrast to the most recent LDEO climatologies where trends over distinct time periods were investigated and one trend then selected for use in time-normalization throughout much of the global ocean (i.e., T-2002, T-2009, and T-2014).

Atmospheric $pCO_2$ change drives rising ocean $pCO_2$, and surface ocean carbon concentrations follow atmospheric increases on multi-decadal timescales and over large regions and the global scale (Fay & McKinley 2013, McKinley et al. 2020). Large synthesis efforts by those in the global ocean carbon community show that even if ocean trends are larger/smaller than the atmosphere on decadal or multi-year time
periods, when considering the longest time periods, the atmosphere and ocean carbon trends are statistically indistinguishable over much of the global ocean (Fay & McKinley 2013, Tjiputra et al. 2014). While the Tjiputra et al (2014) and Fay & McKinley (2013) efforts consider large regional analysis, Bates et al. (2014) provides a synthesis of trends in $pCO_2$ at long-term observing stations, with most of the stations showing a
match to the rise in atmospheric $CO_2$ concentration (Tanhua et al. 2015).

Supplementary Table 1 shows biome-scale mean $fCO_2$ trends computed using all available 1º×1º grid cells with observations in the gridded product released as part of SOCATv2022 (Sabine et al. 2013). We present seasonal trends for each biome due to
seasonal sampling bias over much of the global oceans. Similar to T-2009 (see Tables 1-5 of T-2009), trends for different ocean regions vary significantly (Supplementary Table 1) due in part to differing years with available data across ocean regions (Supplementary Figure 5). Observations within the Indian Ocean, for example, and other regions with blue shading in Supplementary Figure 5 are weighted towards the
1980's and 1990's while regions stretching across the North Atlantic and North Pacific have been heavily sampled over the last two decades with ships of opportunity; These well sampled regions have median years of collected observations later than 2010 (areas with red shading in Supplementary Figure 5).

Recent studies (Friedlingstein et al. 2022) demonstrate that globally, the oceans lag slightly behind the atmosphere in terms of rates of carbon increase, and thus $\Delta fCO_2$ has



become increasingly negative as noted above. The central climatological year represented by our method is thus somewhat ambiguous regionally, though globally it is centered at about 2010; the median year of all $fCO_2$ observations collected in
SOCATv2022 greater than 100km from land is 2013; since observations are more densely clustered in the recent period, observations in the early period may have a greater weight in determining climatological values. Given global trends, our approach may estimate a smaller ocean sink in regions where the ocean was sampled more heavily early in the time period (e.g., blue shading in Supplementary Figure 5) and a
greater ocean ocean sink in regions with heavy recent sampling (red shading in Supplementary Figure 5). We acknowledge that the assumption of a constant $\Delta fCO_2$ does not take into account the long-term trend in $\Delta fCO_2$, however, our simplified approach avoids application of trends determined for well-observed regions and time periods across poorly-observed regions and time periods.


We conducted a sensitivity analysis to demonstrate the impact of the $\Delta fCO_2$ method implemented in this version versus a normalization approach similar to that applied in T-2009. Specifically, we assumed a homogeneous 1.5 µatm $yr^{-1}$ trend in $fCO_2^{oce}$ for all regions and years, and normalized available observations to a reference year of 2010.
Spatial maps of the differences in $fCO_2$ for the year 2010 for the normalization approach versus $\Delta fCO_2$ method are shown in the supplemental information (Supplementary Figure 6). Globally, the annual ocean uptake created using the 1.5 µatm $yr^{-1}$ normalization method (T-2009) is within 3% of the $\Delta fCO_2$ method (this analysis); specifically, -1.85 PgC $yr^{-1}$ versus -1.79 PgC $yr^{-1}$ for the 1.5 µatm $yr^{-1}$ normalization
method and the presented $\Delta fCO_2$ approach, respectively, for a reference year of 2010.

## 3.2 Method for time–space interpolation

The method for spatial interpolation and day of year utilized in the climatology has not changed from the previous versions (e.g., T-2009). As described above, $\Delta fCO_2$ values
are used to compile observations into one reference year in contrast to the time-normalization approach of T-2009. The spatial interpolation scheme requires that all observations are binned into 4º×5º grids for each day of the year. In some areas of the global ocean, such as the northern and equatorial ocean regions, there are observations in a majority of the pixels. However, vast expanses in the Southern
Hemisphere have few observations in each pixel and there are many pixels that contain no observations at all (Figure 1).

We follow the same methodology as T-2009 for binning observations in sparsely sampled grid cells south of 12ºS. Here, spatial binning is increased by 4º and 5º longitude and latitude, respectively, extending from the center of each grid cell. This





creates a grid of overlapping 8°x10° grid cells. Additionally, bins include the day before and after a given day of year. The mean is computed by weighting measured values inversely proportional to their time-space distance from the pixel center. After the above procedures are applied, more than 50% of the space–time pixels over the global oceans
are filled.

To estimate the $\Delta fCO_2$ values in the remaining cells, an interpolation equation based on the 2-D diffusion–advection transport of surface waters is used, as in T-2009. The equation is discretized onto a 4°x5° grid for the global ocean, and solved iteratively
using a finite difference algorithm (Takahashi et al. 1995, T-1997). The method avoids singularities at the poles by assigning land to each high latitude region (Antarctica in the south and treating the highest latitudes of the Arctic Ocean as land in the north). With this method, the resulting $\Delta fCO_2$ values are the solutions obtained after 500 iterations, as previously determined on the basis of interpolation experiments of temperature
values (T-2009).

With this interpolation scheme, observed $\Delta fCO_2$ values where available are preserved, and the continuity equation is used to compute values for grid cells that have no observations. Consistent with previous iterations of this approach, the combined effects
of internal sources and sinks of carbon, $CO_2$ exchange with the atmosphere, as well as upwelling of deep waters are all assumed to be included in the analysis of the observations that feed into the interpolation scheme. Uncertainties persist due to the sparsity of input data, normalization to a reference year, and the space-time interpolation. In part to address these uncertainties, we report only monthly means.

To maintain consistency with similar products and input datasets for flux calculations, we downscale to one degree boxes by assigning all 20 1°x1° pixels in a 4°x5° grid cell the same $\Delta fCO_2$ value. When calculating sea-air fluxes, because the other inputs to the flux calculation such as wind speed, are varying on a 1°x1° degree resolution grid,
differences in the gridded flux climatology emerge on this finer spatial scale.

## 3.3 Flux calculation method: pySeaFlux

To assess the near-global ocean carbon sink associated with these $\Delta fCO_2$ estimates, air-sea $CO_2$ exchange must be calculated. The gridded monthly 1°x1° $\Delta fCO_2$ values
were used to compute air-sea $CO_2$ fluxes using the bulk formulation with python package Seaflux.1.3.1 (https://github.com/lukegre/SeaFlux, Gregor & Fay, 2021). The net sea–air $CO_2$ flux (F) is estimated using:

$$\text{Flux}=k_w \cdot \text{sol} \cdot (fCO_2^{oce}-fCO_2^{atm}) \cdot (1-\text{ice}) \qquad\qquad \text{Eq. (1)}$$



where kw is the gas transfer velocity, sol is the solubility of $CO_2$ in seawater (in units of mol $m^{-3}$ $\mu atm^{-1}$), $fCO_2^{oce}$ is the partial pressure carbon in the surface ocean (in $\mu atm$), and $fCO_2^{atm}$ (in units of $\mu atm$) represents the atmospheric $CO_2$ levels in the marine boundary layer. Finally, to account for the seasonal ice cover at high latitudes, the

fluxes are weighted by one minus the ice fraction (ice), i.e., the open ocean fraction. By utilizing the pySeaFlux package (Fay & Gregor et al. 2021, Gregor & Fay 2021), we are able to include multiple scaled gas transfer velocities for three different wind products and our resulting flux estimate is a mean of the three. Additional inputs to the flux calculation include EN4.2.2 salinity (Good et al. 2013), SST and ice fraction from NOAA

Optimum Interpolation Sea Surface Temperature V2 (OISSTv2, Reynolds et al., 2002), European Centre for Medium-Range Weather Forecasts (ECMWF) ERA5 sea level pressure (Hersbach et al. 2020). Finally, surface winds and associated wind scaling factor for the Cross-Calibrated Multi-Platform v2 (CCMP2; Atlas et al. 2011), the Japanese 55-year Reanalysis (JRA- 55; Kobayashi et al. 2015), and the ECMWF ERA5

(Hersbach et al. 2020) reanalysis products are used.

Fluxes reported here use inputs from the year 2010 for the kw, sol, and ice fraction variables. Alternatively, we have calculated fluxes using a mean over a 17-year time period centered on the year 2010. This yields a very similar value with the mean

difference being a 0.04 PgC $yr^{-1}$ increase in estimated carbon uptake.

# 4. Results

## 4.1 Climatological mean distribution of surface water $\Delta fCO_2$

### 4.1.1 Global

The near-global 12-month climatological mean distribution of $\Delta fCO_2$ ($fCO_2^{oce}$ minus

$fCO_2^{air}$) is reported for the SOCAT database (Figure 2, Fay et al. 2023). Evident in the mapped climatology are the large-scale patterns across the global ocean: the consistent high (positive) $\Delta fCO_2$ values in the equatorial Pacific region where upwelling is a dominant influence, and low (negative) values of $\Delta fCO_2$ in the North Atlantic region where evaporation leads to increase salinity and cooling driving strong uptake of carbon

and subduction of surface waters.

A near-global mean climatology curve shows a bimodal shape in $\Delta fCO_2$, with a smaller peak in boreal spring (March/April) and a much larger peak in late boreal summer (August/September). The curve reaches its minimum in November and begins a

recovery throughout the boreal winter before dipping again to a springtime minimum in



June. The near-global annual mean $\Delta fCO_2$ value is -4.1µatm and it is notable that the mean $\Delta fCO_2$ value is below zero for every month of the year, suggesting that seasonally the global ocean mean is a perpetual carbon sink with expansive regions of uptake nearly always outweighing smaller regions of efflux (Figure 2, 3).


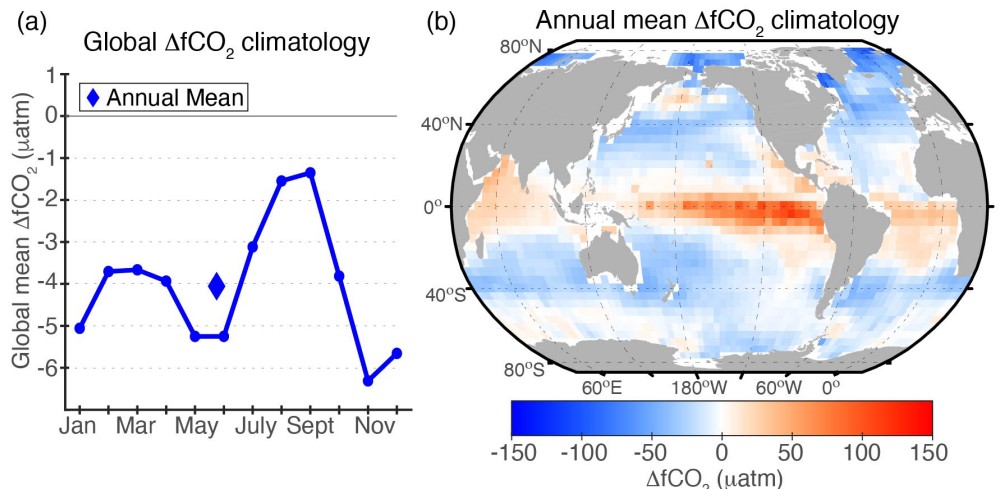

Figure 2: (a) Global mean $\Delta fCO_2$ seasonal climatology from the SOCAT database; annual mean value is indicated by the diamond (-4.1µatm). (b) Map of annual $\Delta fCO_2$ climatology.


Figure 3: Monthly mean values for sea–air ΔfCO₂. Warm colors indicate positive ΔfCO₂ (ocean is greater than atmospheric $CO_2$), white indicates near zero ΔfCO₂, and cool colors indicate negative ΔfCO₂ (ocean $CO_2$ is lower than the atmosphere).



### 4.1.2 Regional

To show the seasonal changes in $\Delta fCO_2$ more clearly it is valuable to consider the patterns exhibited over consistent biogeochemical regions around the globe. For this analysis we utilize the biomes of Fay & McKinley (2014), but for simplicity we combine the seasonally stratified and permanently stratified subtropical biomes into one region in the Northern Hemisphere (referred to simply as subtropical in this manuscript). Monthly climatologies for each of the biomes are shown (Figure 4) in addition to the gridscale maps for each climatological month which allows for further regional interpretation (Figure 3).

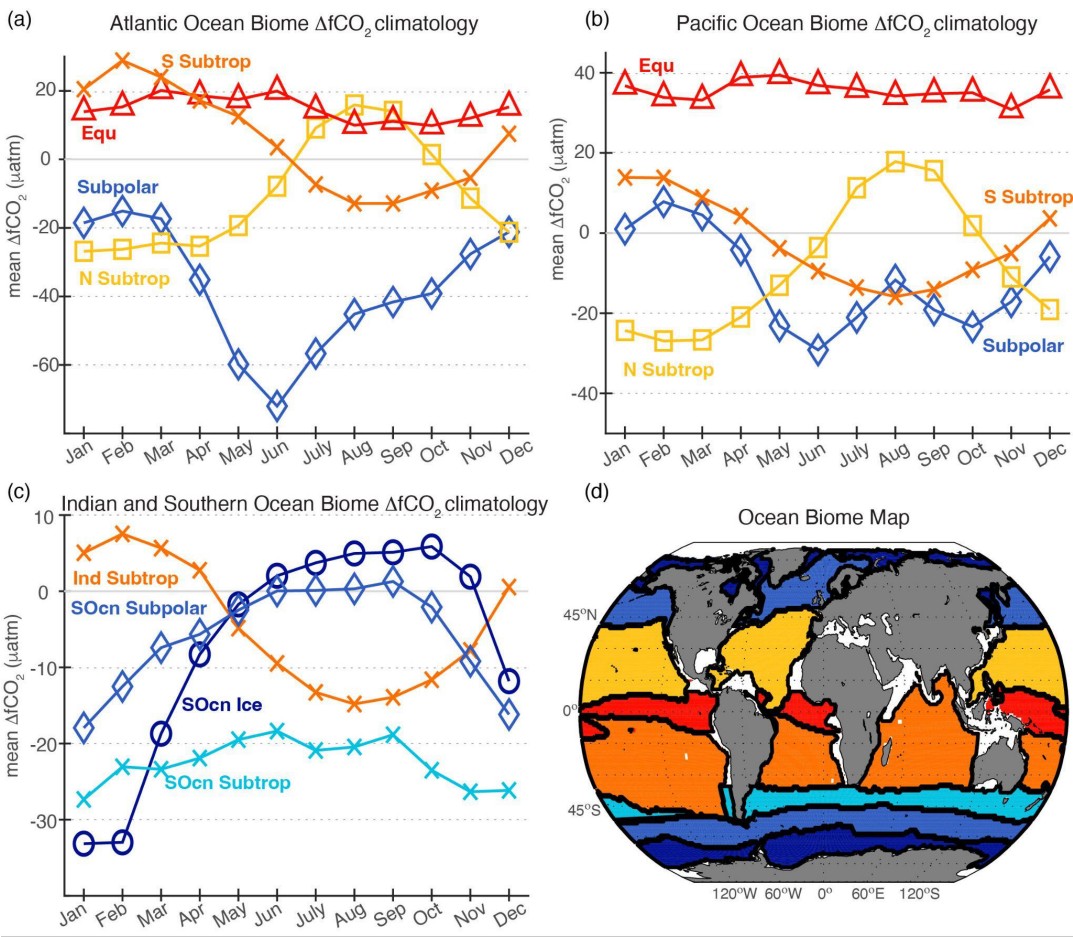

Figure 4: Monthly climatology of $\Delta fCO_2$ for each regional ocean biome in the (a) Atlantic, (b) Pacific, (c) Indian and Southern Ocean basins. (d) Map of regional biomes. Colors of curves correspond to regions on the map in (d) with labels in matching colored text. Note that the y-axis varies between subplots.




The equatorial regions of the Pacific and Atlantic oceans have positive $\Delta fCO_2$ values throughout the annual cycle and little seasonal variability. This indicates that the area is a source of $CO_2$ to the atmosphere year round. The equatorial Pacific (Figure 4b) has
the highest positive $\Delta fCO_2$ values (annual mean = 35.4 µatm), followed by the tropical Atlantic (Figure 4a, annual mean = 14.8 µatm).

The subtropical biomes, representing the temperate North and South Atlantic and Pacific basins exhibit large seasonal $\Delta fCO_2$ cycles which change sign throughout the
year. Here, the $\Delta fCO_2$ cycle is largely temperature driven; positive $\Delta fCO_2$ in warm summer months and negative values in colder winter months reflecting the dominance of seasonal temperature changes on the cycles of ocean $fCO_2$ in these regions. The seasonal amplitude for the subtropical North Pacific is 44.7 µatm, and is slightly larger than the seasonal amplitude in the subtropical North Atlantic (42.7 µatm). Since the
mean seasonal amplitudes for SST are quite similar in these two ocean basins, with the Atlantic having a slightly larger seasonal change in surface temperature (4.4ºC in Pacific and 5.0ºC in Atlantic, not shown), the difference in $\Delta fCO_2$ amplitudes between the Pacific and Atlantic subtropical regions cannot be attributed solely to SST, and may reflect differences in biogeochemical cycling between these two basins.

Seasonal changes in the northern subtropical oceans are roughly six months out of phase from the southern subtropical biomes. The South Pacific subtropical biome has a seasonal amplitude of 29.8 µatm which is nearly 15 µatm smaller than that of the North Pacific subtropical biome. In contrast, the seasonal amplitude of the South Atlantic
subtropical basin is just 1 µatm smaller than its counterpart in the North Atlantic (the South Atlantic subtropical amplitude is 41.7 µatm). The Indian Ocean subtropical biome which encompasses most of the Indian Ocean, both above and below the Equator, has a smaller $\Delta fCO_2$ amplitude (22.3 µatm), but the phasing matches well with both the South Pacific and South Atlantic subtropical biomes, with peak (positive) $\Delta fCO_2$ values
in February and the lowest values in August. The smaller $\Delta fCO_2$ seasonal amplitudes in the Indian and South Pacific subtropical basins are partially attributable to lower SST variability in these regions (SST seasonal cycle amplitudes are 4.0ºC and 3.0ºC in the South Pacific and Indian subtropics, respectively, compared to 4.6ºC in the subtropical South Atlantic). However, it is likely that both differences in spatiotemporal patterns of
primary productivity and undersampling in the South Pacific and Indian subtropics (Figure 1) also contribute to differences in $\Delta fCO_2$ seasonal amplitudes between these basins.





The timing of the $\Delta fCO_2$ trough in the subpolar regions is opposite that of the subtropical
North Pacific and Atlantic basins. Strongly negative $\Delta fCO_2$ in the spring and summer
months is due to the effects of intense biological drawdown which quickly and
dramatically lowers carbon levels in the subpolar surface ocean with the onset of the
growing season. Biological productivity and strong spring/summer stratification result in
subpolar seasonal cycles that are roughly four to six months out of phase compared to
adjacent subtropical regions. In the Atlantic subpolar biome, $\Delta fCO_2$ values are
consistently below zero throughout the annual cycle (a maximum of -15.1 µatm occurs
in February). In the subpolar North Pacific basin, positive $\Delta fCO_2$ values are present over
the boreal winter (Jan-March) before biological drawdown associated with the spring
bloom lowers the $\Delta fCO_2$ values back below zero for the remainder of the year. The
spring drawdown is weaker in the subpolar North Pacific compared to the subpolar
North Atlantic.

Figure 4c displays the seasonal cycle for the Southern Ocean biomes including the
seasonal ice biome, the subpolar region, and the seasonally stratified subtropical region
of the Southern Hemisphere. The higher latitude subtropical region has negative $\Delta fCO_2$
values throughout the year and a relatively small seasonal $\Delta fCO_2$ amplitude compared
to the more expansive South Atlantic, South Pacific and Indian subtropical basins to the
north. The mean $\Delta fCO_2$ and seasonal amplitude for the Southern Ocean subtropical
region is -22.5 µatm and 9.0 µatm, respectively.
The Southern Ocean subpolar and ice biomes both have relatively strong seasonal
cycles, reaching maximums of $\Delta fCO_2$ near and slightly above zero, respectively, during
the late austral winter and early austral spring (Figure 4c). This positive peak during July
through October in the Southern Ocean seasonal ice zone is influenced by under-ice
vertical mixing with deep waters that contain excess carbon and nutrients. During the
austral spring and summer months, intense phytoplankton blooms occur near and
around the edges of the retreating sea ice in the seasonal ice zone and within the
subpolar region. These blooms cause dramatic drops in $\Delta fCO_2$ values at the end of the
calendar year (Oct-Dec). Limited sampling and smoothing from the interpolation method
fail to capture the high spatiotemporal variability that characterizes this highly dynamic
region.

## 4.2 Net air-sea CO$_2$ flux

The mean climatological global air-sea $CO_2$ flux estimate using the SOCAT database is
-1.79 PgC yr$^{-1}$, indicating uptake of carbon by the ocean. This is a slightly greater flux
into the ocean than the direct estimate from the previous version of the climatology (T-
2009), which reported a direct estimated global mean flux of -1.4 PgC yr$^{-1}$ for the year
2000. For the uncertainty in global ocean-atmosphere $CO_2$ flux, we use the value




reported by Wanninkhof et al. (2013) who followed the same approach as T-2009. This approach combines uncertainty contributions from the spatial and temporal sampling of

$\Delta fCO_2$ (±0.18 and ±0.5 PgC yr$^{-1}$, respectively) as well as smaller contributions for the uncertainty in the gas exchange parameterization (±0.2 PgC yr$^{-1}$), wind (±0.15 PgC yr$^{-1}$) and riverine carbon (±0.2 PgC yr$^{-1}$).

## 4.2.1 Mean annual distribution

The near-global mean flux estimate presented here represents 90% of the surface area

of the global ocean. Specifically, this estimate excludes the coastal ocean and areas of the high latitude seas. We have chosen to present values without any adjustment to account for missing areas and acknowledge that this analysis represents a near-global estimate. Suggested methods for filling missing ocean areas in such reconstructions are available in Fay & Gregor et al. (2021) but are not implemented in this climatology to

remain consistent with its previous versions. An estimate of the flux from regions missing from this product can be obtained by using the full-coverage $pCO_2$ climatology combining open and coastal oceans (Landschützer et al. 2020). Considering only grid cells in the Landschützer et al. (2020) product that are missing in this climatology, we estimate an annual average coastal and high latitude flux of -0.38 PgC yr$^{-1}$. The flux

varies through the seasonal cycle, ranging from -0.43 to -0.31 PgC yr$^{-1}$. This quantity is not included in the climatological estimate presented here.

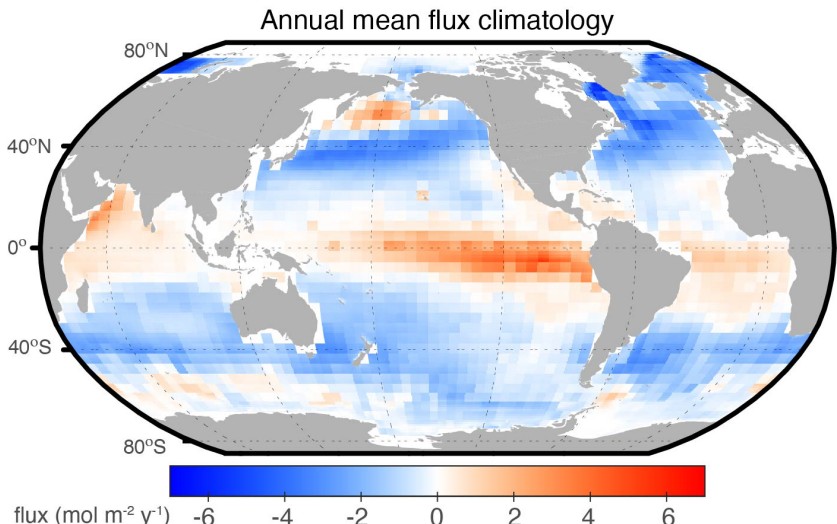

Figure 5: Annual mean $CO_2$ flux calculated from the SOCAT database. Flux is

calculated using the SeaFlux method using the mean of three wind speed reanalysis products. Warm and cool colors indicate regions of carbon efflux and uptake, respectively. The near-global mean flux is -1.79 PgC yr$^{-1}$.



Figure 5 shows the climatological mean annual sea–air $CO_2$ flux (mol m$^{-2}$ yr$^{-1}$) and maps of two seasons (DJF and JJA) are displayed in Figure 6. The equatorial Pacific is the most prominent atmospheric $CO_2$ source region, with a seasonally persistent sea-to-air flux of 0.35 PgC yr$^{-1}$. When combined with the equatorial Atlantic region, the tropical belt emits an annual mean of 0.39 PgC yr$^{-1}$ to the atmosphere. Adjacent to this tropical

efflux zone, are vast expanses of seasonally-variable flux patterns. The subtropical basins in both hemispheres act as $CO_2$ sinks in the cooler months and transition to regions of neutral or small $CO_2$ sources during the warmer months. At higher subtropical latitudes, strong winds and relatively low ocean $fCO_2$ occur along the subtropical convergence zone where the cooled subtropical gyre waters with low $fCO_2$ meet the

subpolar waters with biologically-lowered $fCO_2$.

The Northern Hemisphere mid and high latitude regions represent a smaller sink (-0.63 PgC yr$^{-1}$) compared to the corresponding regions of the Southern Hemisphere (-0.91 PgC yr$^{-1}$) largely due to the overall greater surface area of the oceans in the Southern

Hemisphere (oceans south of 35S are 25% of total global ocean area while oceans north of 35N are 15% of total ocean area). The dramatic influence of the expansive Southern Hemisphere oceans is also demonstrated by the large flux in the Southern Ocean subtropical region (-0.59 PgC yr$^{-1}$) that represents 8% of the global ocean surface area.


Moving poleward, a strong sink (-0.27 PgC yr$^{-1}$) occurs in the North Atlantic subpolar region which includes the Nordic Seas and the portions of the Arctic Ocean which contain observations. This strong localized carbon sink is attributed to the import of low anthropogenic waters at depth in the Gulf Stream that are exposed as mixed layers

deepen (Ridge & McKinley 2020), and large phytoplankton blooms in spring followed by cooling in winter. In the Southern Ocean, annual mean $CO_2$ flux is heterogeneous and relatively small in the seasonal ice zone due to the ice cover that reduces sea–air gas transfer in winter. Additionally, the small annual flux values in the Southern Ocean subpolar and ice regions (-0.21 PgC yr$^{-1}$ and -0.08 PgC yr$^{-1}$, respectively) are a result of

a cancellation of the seasonal source (winter) and sink (summer) fluxes.

### 4.2.2 Seasonal variation of sea–air $CO_2$ flux

Seasonal variation in air-sea fluxes are clearly seen in the climatology (Figure 6) and are attributed to a combination of effects including fluctuations in SSTs, biological

uptake of carbon dioxide, as well as mixing and wind speeds.





The seasonal variability of fluxes in higher latitudes of the subtropics in the Atlantic, Indian and Pacific Oceans cause an oscillation from neutral or weak sources of $CO_2$ to the atmosphere in the warmer seasons to strong $CO_2$ sinks in the cooler or winter

seasons. Water cools as it is transported poleward by western boundary currents, allowing for carbon uptake (Ayers & Lozier 2012). In spring and summer, the biological drawdown of carbon increases $CO_2$ uptake by the ocean, partially offset by increases in $fCO_2$ due to warming.

Subtropical gyre regions also transition from weak sinks in the winter seasons to weak sources in the summer seasons, following the seasonal SST cycles and reflecting the dominance of temperature effects in controlling the seasonal variability in the $fCO_2$ and sea–air fluxes in oligotrophic gyres. In tropical low latitude regions, seasonality is generally smaller, however localized hot spots of high variability and large fluxes do

exist, such as in the northwestern Indian Ocean where the strong summer monsoon winds force upwelling of carbon rich subsurface waters and cause high gas transfer rates in this region (Chen et al. 1998, T-2002). The equatorial Pacific and Atlantic show little seasonal variability in $CO_2$ flux with a persistent efflux throughout the year.

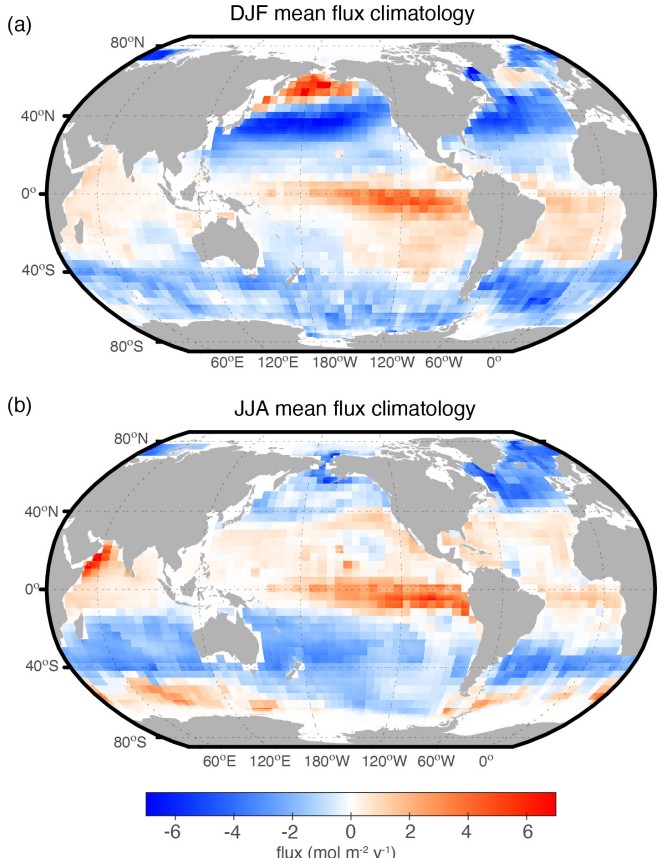

Figure 6: Season sea–air CO$_2$ flux (molC m$^{-2}$ year$^{-1}$) climatology for (a) December, January, February (DJF) and (b) June, July, August (JJA). Positive values (warm colors) indicate sea-to-air fluxes (ocean efflux), and negative values (cool colors) indicate air-to-sea fluxes (ocean uptake).

In the Southern Ocean, there is a consistent region of moderate carbon source waters located in the Atlantic and Indian sector south of 45S latitude during the austral winter (Figure 6b). The source in the Southern Ocean region could be influenced by high fCO$_2$ waters from margins of the Antarctic sea-ice field given that the efflux values occur during the austral winter months (JJA). As the seasons transition to warmer temperatures and the ice edge recedes, this region is impacted by high rates of photosynthesis causing fCO$_2$ drawdown, and resulting in a transition to moderate carbon sinks during the austral summer (DJF). Uncertainties are higher in the Southern Ocean region due to the limited number of observations, particularly in winter (Figure 1).



# 5. Discussion

It should be noted that there isn't one specific reference year for this release of the $\Delta fCO_2$ climatology as was the case for previous releases (e.g., the year 2000 reference in T-2009). Instead, this climatology represents a multidecadal time period, beginning in 1980, with the majority of observations feeding into the climatology collected after 2010. Therefore, while the $\Delta fCO_2$ climatology is not reported for a specific reference year, it is

most representative of the conditions over the past two decades. We note however that the flux estimates given in this analysis are based on inputs from a single year, the year 2010, as described above in Section 3.3 (comparison of flux estimates using 2010 inputs and averages over several decades yield very similar results, also as described in Section 3.3).


## 5.1 Comparison with T-2009 climatology

In the previous release of this climatology (T-2009), $pCO_2$ values were corrected to a reference year of 2000 using a mean atmospheric $CO_2$ increase rate of 1.5 µatm yr$^{-1}$

over the entire ocean area with the exception of the Bering Sea, where the observed rate of -1.2 µatm yr$^{-1}$ was used. In our updated method described above (Section 3.1), we eliminate the need to apply a normalization rate for observations and instead calculate a $\Delta fCO_2$ value for each observation using a co-located concurrent atmospheric $fCO_2$ value. We note that T-2014 presents a more updated $pCO_2$ climatology than T-

2009; since T-2014 emphasizes climatologies for the other carbonate system variables (pH, dissolved inorganic carbon, and total alkalinity) and omits estimation of fluxes, we focus our comparison on values presented in T-2009.

Spatial differences between the climatology created from surface water $\Delta fCO_2$ values

using the methods discussed above and the approach of T-2009 (3 million observations) are shown in Figure 7 for months February and August; months were selected for consistency with past comparisons presented in T-2009 and T-2002. The differences between this updated release and previous versions producing climatologies for reference years 2000 (T-2009) and 1990 (T-1997) are unlikely to represent real

change in the oceans over time, but instead primarily reflect the impact of the greatly expanded database as well as the use of the $\Delta fCO_2$ approach as opposed to the time normalization method of T-2009.

The most significant regional differences between this updated version and that of T-

2009 are observed over the subpolar North Atlantic, the subtropical Southeast Pacific and portions of the Southern Ocean. In the North Atlantic, differences between versions





are largest in the boreal winter (February map, top panel of Figure 7) when the updated climatology exhibits less uptake compared to the T-2009 version (positive values on the map indicate more negative values in the T-2009 version). Differences in the North

Atlantic can be at least partially attributed to the much greater availability of observations in this region between the two databases (Figure 1, Supplementary Figure 1). This is discussed in further detail in the supplementary text.

        The Southeast Pacific is an area with very limited observations but where a few key
datasets have been included in the SOCAT database since 2010. Figure 1 shows that despite these recent additions to SOCAT, there are still only a few observations covering this region. Comparison of Figure 3 in this study with Figure 9 of T-2009 suggests that the additional datasets in SOCATv2022 result in a more defined seasonal cycle for $fCO_2$ in the subtropical Southeast Pacific in the current release. Specifically,
the map of differences shown in Figure 7 shows that this region is a greater source of carbon to the atmosphere in austral summer and a greater sink in austral winter compared to T-2009 (compare also Figure 6 of this study with Figure 15 of T-2009). In contrast, monthly maps included in Figure 3 of T-2009 show little seasonal contrast in $\Delta pCO_2$ likely due to a lack of observations.
        In the Southern Ocean region during austral winter (August), $\Delta fCO_2$ values are more negative in the current version compared to T-2009. The Southern Ocean is also a region of limited data availability particularly in austral winter but one where SOCATv2022 also includes several datasets added in the past decade that have an
outsized influence on the resulting climatology.


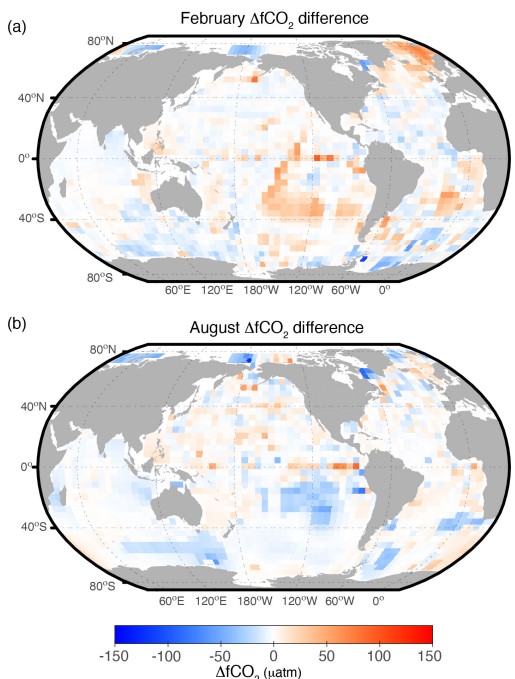

Figure 7: Difference maps for the surface water $\Delta fCO_2$ climatology produced by this study using SOCAT and that from T-2009 (maps show this study *minus* T-2009) in (a) February and (b) August. In T-2009, the delta $CO_2$ values are reported as $pCO_2$ and here we are using $fCO_2$.

## 5.2 Comparison to other flux estimates

The estimate presented here for annual global mean carbon flux (-1.79 PgC yr$^{-1}$) represents slightly less uptake than reported by other studies, but given uncertainties, as well as differing timeframes, spatial coverage and gap-filling methodologies, our estimate compares closely with current estimates similarly based on observed surface ocean $fCO_2$.

To compare our new climatological estimate of contemporary air–sea net flux from surface ocean $fCO_2$ with estimates of the anthropogenic carbon flux from interior data (e.g., Gruber et al. 2019) or estimates from global ocean biogeochemical models (e.g., Friedlingstein et al. 2022; Hauck et al. 2020), it is necessary to account for the outgassing of natural carbon supplied to the ocean by rivers. This riverine estimate varies significantly in magnitude between studies and continues to be a research focus for the ocean carbon community. Therefore, we focus on comparisons between our



climatological estimate and a mean carbon flux estimate from an ensemble of observation-based $pCO_2$ products included in the SeaFlux product (Fay & Gregor et al. 2021).


The SeaFlux products span the years 1985-2020, are all similarly based on the SOCAT database, but employ various methods of machine learning and interpolation to produce full coverage ocean carbon maps. For this comparison, a climatology of the SeaFlux product is produced and fluxes are calculated in the same manner as for the climatology presented here. Following this approach, the SeaFlux climatology ensemble yields a global mean flux of -2.10 PgC yr$^{-1}$ which represents a slightly larger flux into the ocean than that produced by the updated climatology (-1.79 PgC yr$^{-1}$). The differences in global flux can be attributed to the true global coverage of the SeaFlux product relative to the 90% global ocean coverage of this study. As mentioned above in Section 4.2, an estimate of missing coastal and high latitude fluxes increases the ocean carbon uptake estimate for this climatology by roughly 0.38 PgC yr$^{-1}$. Adding this additional flux brings our analysis within 0.1 PgC yr$^{-1}$ of the SeaFlux ensemble (-2.17 PgC yr$^{-1}$ versus -2.10 PgC yr$^{-1}$ for this analysis versus the SeaFlux estimate, respectively).




Spatially, comparison of the SeaFlux ensemble of products to our climatology shows strong agreement in overall patterns but significant differences in the mid and high latitude Southern Hemisphere oceans (Figure 8). Gloege et al. (2021) analyzed a machine learning method's ability to reconstruct global carbon fluxes from available observations using a testbed approach and found the highest flux bias in the Southern Ocean regions as well as an overestimation of decadal variability in this region. Given the limited availability of year-round observations at high Southern Hemisphere latitudes, and the resulting reliance on various gap-filling approaches, it is not surprising that the largest differences between the climatology presented here and the SeaFlux ensemble emerges in this region. Significant differences between these climatologies are also evident in the high latitude North Pacific and North Atlantic, specifically in the boreal winter season (Figure 8a). Again, a lack of observations in these regions during the winter season (Figure 1) likely accounts for much of this disagreement, with more reliance on the interpolation methods used by each method. Machine learning methods that utilize proxy variables to estimate $pCO_2$ in unsampled areas, such as those in the SeaFlux product, often rely on relationships derived from better-observed areas that are deemed similar in biogeochemical characteristics and it is likely that the mechanisms at these high latitude locations are not accurately captured by any available interpolation methods. This is also a current focus of research for the ocean carbon observing community.









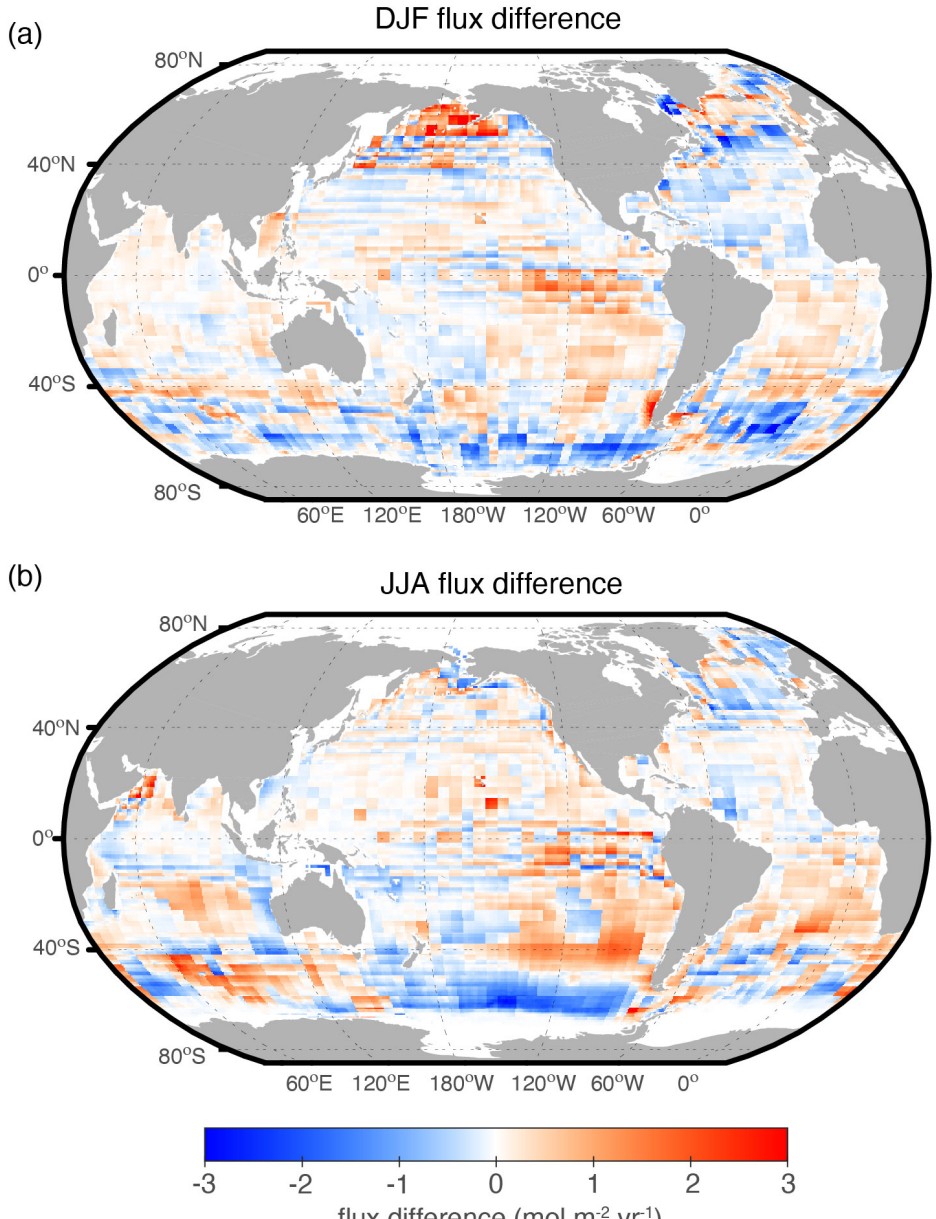

Figure 8: Difference map for carbon fluxes (mol m⁻² yr⁻¹) calculated from this ΔfCO₂
climatology and fluxes reported by an ensemble of observation-based products included
in the SeaFlux product for (a) boreal winter, DJF and (b) boreal summer, JJA. Map
shows the difference defined as this study *minus* SeaFlux.

# 6. Conclusion

An updated climatological mean distribution for $\Delta fCO_2$ (surface water minus
atmosphere) using the methods of T-2009 is presented. This climatology is based on
approximately seven times more open ocean observations from the SOCATv2022
database (over 21 million values, spanning years 1980-2021) compared to the 3 million
data values used in T-2009 (and more than three times the approximately 6.5 million
observations used in T-2014). In this updated climatology, observations made during El
Niño periods over the equatorial Pacific are included, unlike climatologies presented by
T-1997, T-2002, T-2009 and T-2014. In addition to coastal waters, the highest latitudes
of the Arctic and the Mediterranean Sea are also excluded as in all previous LDEO
climatologies.

To develop a climatology from data collected over multiple decades during which $fCO_2$
experienced a large secular trend, we calculate $\Delta fCO_2$ values for each day and grid cell
before collapsing all available data onto one climatological year. This method follows
the assumption made in previous iterations of this climatology (T-1997) that the ocean
surface carbon value follows the rate of increase in the atmospheric $fCO_2,$ such that
$\Delta fCO_2$ is constant over time. Observed $\Delta fCO_2$ is then interpolated in space-time  using a
lateral two-dimensional diffusion–advection equation on a 4º×5º grid (Takahashi et al.
1995, T-1997, T-2002, T-2009, T-2014). Monthly mean $\Delta fCO_2$ values for each pixel,
downscaled to 1º×1º resolution, are presented here. Net sea-air $CO_2$ flux is computed
using the pySeaFlux package, following the protocol presented in Fay & Gregor et al.
(2021).

Regional mean $\Delta fCO_2$ values vary greatly among the ocean basins (Figures 3 and 4).
The high-latitude North Atlantic is the most intense $CO_2$ sink per unit area as a result of
both highly negative $\Delta fCO_2$ (Figure 3) and strong winds. This is also the region with the
largest differences between the climatologies created with previous versions of the
LDEO database and this version based on the SOCATv2022 database (Figures 3 and
7). Globally, differences are due to the greater abundance of observations over all
regions of the global oceans in the SOCAT database, but particularly the greater
seasonal coverage in the Southern Hemisphere oceans and subpolar North Atlantic
(Figure 1, Supplementary Figure 1).

The annual mean uptake flux for the global open-ocean region is estimated to be -1.79
PgC yr$^{-1}$ for 1980-2021 (Figures 5 and 6). Of the major ocean basins, the Southern
Hemisphere oceans are the largest $CO_2$ sink, taking up 1.19 PgC yr$^{-1}$, while the
Northern Hemisphere oceans (subtropical and subpolar biomes) take up 1.04 PgC yr$^{-1}$.



The equatorial oceans act as the only year-round region of carbon efflux to the atmosphere with and emit 0.35 PgC yr$^{-1}$ to the atmosphere.

While over a million new shipboard fCO$_2^{oce}$ observations have been made each year in the global oceans for the past two decades, there has been a notable decline in the observations submitted to SOCAT since 2017. This decline is due in part to the disruption of the COVID19 pandemic, but also reflects a shift away from shipboard pCO$_2$ measurements. Given the lack of alternative approaches with which to assess spatial and temporal variability in air-sea CO$_2$ flux and the need for high accuracy shipboard measurements (accuracy of <2 µatm) to characterize most regions of the global oceans, this trend to fewer observations is highly detrimental to carbon cycle research. This is true both in regard to monitoring of ocean carbon uptake and to monitoring of more uncertain fluxes such as that between the atmosphere and terrestrial biosphere since the high uncertainty of independent terrestrial estimates necessitates the monitoring of this flux by difference.

## Data Availability

The updated climatology is available via The National Center for Environmental Information (NCEI) at https://www.ncei.noaa.gov/access/metadata/landing-page/bin/iso?id=gov.noaa.nodc:028225, doi.org/10.25921/295g-sn13, (Fay et al. 2023).

## Author contribution

ARF and DRM conducted the analysis and prepared the manuscript. GAM, RW, CW, SCS, and DP contributed ideas and provided feedback throughout the analysis as well as contributed to manuscript.

## Acknowledgements

The Surface Ocean CO$_2$ Atlas (SOCAT) is an international effort, endorsed by the International Ocean Carbon Coordination Project (IOCCP), the Surface Ocean Lower Atmosphere Study (SOLAS) and the Integrated Marine Biosphere Research (IMBeR) program, to deliver a uniformly quality-controlled surface ocean CO$_2$ database. The many researchers and funding agencies responsible for the collection of data and quality control are thanked for their contributions to SOCAT. Funding was obtained from the NOAA Office of Oceanic and Atmospheric Research (OAR) Global Ocean Monitoring and Observations (GOMO) program.





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

Bakker, D. C. E.; Alin, Simone R.; Becker, Meike; Bittig, Henry C.; Castaño-Primo, Rocío; Feely, Richard A.; Gkritzalis, Thanos; Kadono, Koji; Kozyr, Alex; Lauvset, Siv K.;



Metzl, Nicolas; Munro, David R.; Nakaoka, Shin-ichiro; Nojiri, Yukihiro; O'Brien, Kevin M.; Olsen, Are; Pfeil, Benjamin; Pierrot, Denis; Steinhoff, Tobias; Sullivan, Kevin F.; Sutton, Adrienne J.; Sweeney, Colm; Tilbrook, Bronte; Wada, Chisato; Wanninkhof, Rik; Willstrand Wranne, Anna; Akl, John; Apelthun, Lisa B.; Bates, Nicholas; Beatty, Cory M.; Burger, Eugene F.; Cai, Wei-Jun; Cosca, Catherine E.; Corredor, Jorge E.; Cronin, Margot; Cross, Jessica N.; De Carlo, Eric H.; DeGrandpre, Michael D.; Emerson,
Steven R.; Enright, Matt P.; Enyo, Kazutaka; Evans, Wiley; Frangoulis, Constantin; Fransson, Agneta; García-Ibáñez, Maribel I.; Gehrung, Martina; Giannoudi, Louisa; Glockzin, Michael; Hales, Burke; Howden, Stephan D.; Hunt, Christopher W.; Ibánhez, J. Severino P.; Jones, Steve D.; Kamb, Linus; Körtzinger, Arne; Landa, Camilla S.; Landschützer, Peter; Lefèvre, Nathalie; Lo Monaco, Claire; Macovei, Vlad A.; Maenner

Jones, Stacy; Meinig, Christian; Millero, Frank J.; Monacci, Natalie M.; Mordy, Calvin; Morell, Julio M.; Murata, Akihiko; Musielewicz, Sylvia; Neill, Craig; Newberger, Tim; Nomura, Daiki; Ohman, Mark; Ono, Tsuneo; Passmore, Abe; Petersen, Wilhelm; Petihakis, George; Perivoliotis, Leonidas; Plueddemann, Albert J.; Rehder, Gregor; Reynaud, Thierry; Rodriguez, Carmen; Ross, Andrew C.; Rutgersson, Anna; Sabine,
Christopher L.; Salisbury, Joseph E.; Schlitzer, Reiner; Send, Uwe; Skjelvan, Ingunn; Stamataki, Natalia; Sutherland, Stewart C.; Sweeney, Colm; Tadokoro, Kazuaki; Tanhua, Toste; Telszewski, Maciej; Trull, Tom; Vandemark, Douglas; van Ooijen, Erik; Voynova, Yoana G.; Wang, Hongjie; Weller, Robert A.; Whitehead, Chris; Wilson, Doug (2022). Surface Ocean CO2 Atlas Database Version 2022 (SOCATv2022) (NCEI
Accession 0253659). NOAA National Centers for Environmental Information. Dataset. https://doi.org/10.25921/1h9f-nb73. Accessed July 14, 2022.

Chen, C. T. A., and S. Tsunogai. "Carbon and nutrients in the ocean." Asian Change in the Context of Global Change: 271-307, 1998.

DeVries T., Yamamoto K., Wanninkhof R., Gruber N., Hauck J., Müller J.D., Bopp L.,
Carroll D., Carter B., Chau T., Doney S., Gehlen M., Gloege L., Gregor L., Henson S., Kim J.H., Iida Y., Ilyina T., Landschützer P., Le Quéré C., Munro D., Nissen C., Patara L., Perez F.F., Resplandy L., Rodgers K., Schwinger J., Séférian R., Sicardi V., Terhaar J., Triñanes J., Tsujino H., Watson A., Yasunaka S., Zeng, J.: Magnitude, trends, and variability of the global ocean carbon sink from 1985-2018, submitted to Global
Biogeochemical Cycles, 2023.

Dickson, A. G., Sabine, C. L., and Christian, J. R. (Eds.): Guide to best practices for ocean CO2 measurements, PICES Special Publication 3, IOCCP Report 8, 191 pp., 2007.





845 Fay, A.R. and McKinley, G.A.: Global trends in surface ocean pCO2 from in situ data. Global Biogeochemical Cycles, 27(2), pp.541-557, 2013.

Fay, A. R., & McKinley, G. A.: Global open-ocean biomes: Mean and temporal variability. Earth System Science Data, 6(2), 273–284.
850 https://doi.org/10.5194/essd-6-273-2014, 2014.

Fay, A. R., Gregor, L., Landschützer, P., McKinley, G. A., Gruber, N., Gehlen, M., Iida, Y., Laruelle, G. G., Rödenbeck, C., Roobaert, A., and Zeng, J.: SeaFlux: harmonization of air–sea CO2 fluxes from surface pCO2 data products using a standardized approach,
855 Earth Syst. Sci. Data, 13, 4693–4710, https://doi.org/10.5194/essd-13-4693-2021, 2021

Fay, A. R., Munro, D.R., McKinley, G. A., Pierrot, D., Sutherland, S. C., Sweeney, C., Wanninkhof, R.: Climatological distributions of sea-air Delta fCO2 and CO2 flux densities in the Global Surface Ocean (NCEI Accession 0282251). NOAA National
860 Centers for Environmental Information. Dataset. https://doi.org/10.25921/295g-sn13, 2023.

Feng, S., Lauvaux, T., Keller, K., Davis, K. J., Rayner, P., Oda, T., and Gurney, K. R.: A road map for improving the treatment of uncertainties in high-resolution regional carbon
865 flux inverse estimates, Geophysical Research Letters, 46, 13,461–13,469. https://doi.org/10.1029/2019GL082987, 2019.

Friedlingstein, P., O'Sullivan, M., Jones, M. W., Andrew, R. M., Gregor, L., Hauck, J., Le Quéré, C., Luijkx, I. T., Olsen, A., Peters, G. P., Peters, W., Pongratz, J.,
870 Schwingshackl, C., Sitch, S., Canadell, J. G., Ciais, P., Jackson, R. B., Alin, S. R., Alkama, R., Arneth, A., Arora, V. K., Bates, N. R., Becker, M., Bellouin, N., Bittig, H. C., Bopp, L., Chevallier, F., Chini, L. P., Cronin, M., Evans, W., Falk, S., Feely, R. A., Gasser, T., Gehlen, M., Gkritzalis, T., Gloege, L., Grassi, G., Gruber, N., Gürses, Ö., Harris, I., Hefner, M., Houghton, R. A., Hurtt, G. C., Iida, Y., Ilyina, T., Jain, A. K.,
875 Jersild, A., Kadono, K., Kato, E., Kennedy, D., Klein Goldewijk, K., Knauer, J., Korsbakken, J. I., Landschützer, P., Lefèvre, N., Lindsay, K., Liu, J., Liu, Z., Marland, G., Mayot, N., McGrath, M. J., Metzl, N., Monacci, N. M., Munro, D. R., Nakaoka, S.-I., Niwa, Y., O'Brien, K., Ono, T., Palmer, P. I., Pan, N., Pierrot, D., Pocock, K., Poulter, B., Resplandy, L., Robertson, E., Rödenbeck, C., Rodriguez, C., Rosan, T. M., Schwinger,
880 J., Séférian, R., Shutler, J. D., Skjelvan, I., Steinhoff, T., Sun, Q., Sutton, A. J., Sweeney, C., Takao, S., Tanhua, T., Tans, P. P., Tian, X., Tian, H., Tilbrook, B., Tsujino, H., Tubiello, F., van der Werf, G. R., Walker, A. P., Wanninkhof, R., Whitehead, C., Willstrand Wranne, A., Wright, R., Yuan, W., Yue, C., Yue, X., Zaehle, S., Zeng, J.,





and Zheng, B.: Global Carbon Budget 2022, Earth Syst. Sci. Data, 14, 4811–4900,
https://doi.org/10.5194/essd-14-4811-2022, 2022.

Gloege, L., McKinley, G. A., Landschutzer, P., Fay, A. R., Frol- icher, T., Fyfe, J., Ilyina,
T., Jones, S., Lovenduski, N. S., Röden- beck, C., Rogers, K., Schlunegger, S., and
Takano, Y.: Quantifying errors in observationally-based estimates of ocean carbon sink
variability, Global Biogeochem. Cy., 35, e2020GB006788,
https://doi.org/10.1029/2020GB006788, 2021.

Good, S. A., Martin, M. J., and Rayner, N. A.: EN4: Quality controlled ocean
temperature and salinity pro- files and monthly objective analyses with uncertainty
estimates, J. Geophys. Res.-Oceans, 118, 6704–6716,
https://doi.org/10.1002/2013JC009067, 2013.

Gregor L., & Fay, A. R.: SeaFlux data set: harmonised sea-air CO2 fluxes from surface
pCO2 data products using a standardised approach (2021.04, Data set: Zenodo.
https://doi.org/10.5281/zenodo.5148460, 2021. Last accessed September, 12, 2022.


Gruber, N., Clement, D., Carter, B. R., Feely, R. A., van Heuven, S., Hoppema, M., et
al.: The oceanic sink for anthropogenic co2 from 1994 to 2007. Science 363, 1193–
1199. doi: 10.1126/science.aau5153, 2019.

Gruber, N., Bakker, D. C., DeVries, T., Gregor, L., Hauck, J., Landschützer, P., ... &
Müller, J. D.: Trends and variability in the ocean carbon sink. Nature Reviews Earth &
Environment, 4(2), 119-134, 2023.

Hauck, J., Zeising, M., Le Quéré, C., Gruber, N., Bakker, D. C. E.,Bopp, L., Chau, T. T.
T., Gürses, Ö., Ilyina, T., Landschützer, P., Lenton, A., Resplandy, L., Rödenbeck, C.,
Schwinger, J., and Séférian, R.: Consistency and Challenges in the Ocean Carbon Sink
Estimate for the Global Carbon Budget, Front. Mar. Sci., 7, 852-885,
https://doi.org/10.3389/fmars.2020.571720, 2020.
Hersbach, H., Bell, B., Berrisford, P., Hirahara, S., Horányi, A., Muñoz-Sabater, J.,
Nicolas, J., Peubey, C., Radu, R., Schepers, D., Simmons, A., Soci, C., Abdalla, S.,
Abellan, X., Balsamo, G., Bechtold, P., Biavati, G., Bidlot, J., Bonavita, M., De Chiara,
G., Dahlgren, P., Dee, D., Diamantakis, M., Dragani, R., Flemming, J., Forbes, R.,
Fuentes, M., Geer, A., Haimberger, L., Healy, S., Hogan, R.J., Hólm, E., Janisková, M.,
Keeley, S., Laloyaux, P., Lopez, P., Lupu, C., Radnoti, G., de Rosnay, P., Rozum, I.,
Vamborg, F., Villaume, S., and Thépaut, J.: The ERA5 global reanalysis, Q. J. Roy.
Meteor. Soc. 146, 1999–2049, https://doi.org/10.1002/qj.3803, 2020 (data available at:



https://cds.climate.copernicus.eu/cdsapp#!/dataset/ reanalysis-era5-single-levels-monthly-means?tab=overview, last access: 16 October, 2020).

Kalnay, E., Kanamitsu, M., Kistler, R., Collins, W., Deaven, D., Gandin, L., Iredell, M., Saha, S., White, G.,Woollen, J., Zhu, Y., Chelliah, M., Ebisuzaki, W., Higgins, W., Janowiak, J., Mo, K. C., Ropelewski, C., Wang, J., Leetmaa, A., Reynolds, R., Jenne, R., and Joseph, D.: The NCEP/NCAR 40-year reanalysis project, B. Am. Meteorol. Soc., 77, 437–470, 1996.


Kobayashi, S., Ota, Y., Harada, Y., Ebita, A., Moriya, M., Onoda, H., Onogi, K., Kamahori, H., Kobayashi, C., Endo, H., Miyaoka, K., and Takahashi, K.: The JRA-55 Reanalysis: General Spec- ifications and Basic Characteristics, J. Meteorol. Soc. Jpn., 93, 5–48, https://doi.org/10.2151/jmsj.2015-001, 2015.


Lan, X., Tans, P., Thoning, K., & NOAA Global Monitoring Laboratory. NOAA Greenhouse Gas Marine Boundary Layer Reference - CO2. [Data set]. NOAA GML. https://doi.org/10.15138/DVNP-F961, 2023.

Landschützer, P., Gruber, N. and Bakker, D.C.: Decadal variations and trends of the global ocean carbon sink. Global Biogeochemical Cycles, 30(10), pp.1396-1417, 2016.

Landschützer, P., Laruelle, G. G., Roobaert, A., and Regnier, P.: A uniform pCO2 climatology combining open and coastal oceans, Earth Syst. Sci. Data, 12, 2537–2553,
https://doi.org/10.5194/essd-12-2537-2020, 2020.

Manning, A. C., and Keeling, R. F., Global oceanic and land biotic carbon sinks from the Scripps atmospheric oxygen flask sampling network, Tellus, 58B, pp. 95-116, 2006.

McKinley, G.A., Fay, A.R., Eddebbar, Y.A., Gloege, L. and Lovenduski, N.S.: External forcing explains recent decadal variability of the ocean carbon sink. Agu Advances, 1(2), p.e2019AV000149, 2020.

Pfeil, B., Olsen, A., Bakker, D. C. E., Hankin, S., Koyuk, H., Kozyr, A., Malczyk, J.,
Manke, A., Metzl, N., Sabine, C. L., Akl, J., Alin, S. R., Bates, N., Bellerby, R. G. J., Borges, A., Boutin, J., Brown, P. J., Cai, W.-J., Chavez, F. P., Chen, A., Cosca, C., Fassbender, A. J., Feely, R. A., González-Dávila, M., Goyet, C., Hales, B., Hardman-Mountford, N., Heinze, C., Hood, M., Hoppema, M., Hunt, C. W., Hydes, D., Ishii, M., Johannessen, T., Jones, S. D., Key, R. M., Körtzinger, A., Landschützer, P., Lauvset, S.
K., Lefèvre, N., Lenton, A., Lourantou, A., Merlivat, L., Midorikawa, T., Mintrop, L., Miyazaki, C., Murata, A., Nakadate, A., Nakano, Y., Nakaoka, S., Nojiri, Y., Omar, A.



M., Padin, X. A., Park, G.-H., Paterson, K., Perez, F. F., Pierrot, D., Poisson, A., Rios, A. F., Santana-Casiano, J. M., Salisbury, J., Sarma, V. V. S. S., Schlitzer, R., Schneider, B., Schuster, U., Sieger, R., Skjelvan, I., Steinhoff, T., Suzuki, T., Takahashi, T., Tedesco, K., Telszewski, M., Thomas, H., Tilbrook, B., Tjiputra, J., Vandemark, D., Veness, T., Wanninkhof, R., Watson, A. J., Weiss, R., Wong, C. S., Yoshikawa-Inoue, H.: A uniform, quality controlled Surface Ocean CO2 Atlas (SOCAT) Earth System Science Data , 5, 125-143.doi:10.5194/essd-5-125-2013, 2013.

Quay, P. D., Tilbrook, B., and Wong C. S., Oceanic uptake of fossil fuel $CO_2$: carbon-13 evidence, Science, 256, pp. 74-79, 1992.

Reynolds, R. W., Rayner, N. A., Smith, T. M., Stokes, D. C., and Wang, W.: An improved in situ and satellite SST analysis for cli- mate, J. Climate, 15, 1609–1625, https://doi.org/10.1175/1520- 0442(2002)015<1609:AIISAS>2.0.CO;2, 2002 (data available at: https://psl.noaa.gov/data/gridded/data.noaa.oisst.v2.html, last access: 26 April 2021).

Ridge, S. M., & McKinley, G. A.: Advective Controls on the North Atlantic Anthropogenic Carbon Sink. Global Biogeochemical Cycles, 34(7), 1138. https://doi.org/10.1029/2019gb006457, 2020.

Rödenbeck, C., Bakker, D. C. E., Gruber, N., Iida, Y., Jacobson, A. R., Jones, S., Landschützer, P., Metzl, N., Nakaoka, S., Olsen, A., Park, G.-H., Peylin, P., Rodgers, K. B., Sasse, T. P., Schuster, U., Shutler, J. D., Valsala, V., Wanninkhof, R., and Zeng, J.: Data-based estimates of the ocean carbon sink variability – first results of the Surface Ocean $pCO_2$ Mapping intercomparison (SOCOM), Biogeosciences, 12, 7251–7278, https://doi.org/10.5194/bg-12-7251-2015, 2015.

Sabine, C. L., Hankin, S., Koyuk, H., Bakker, D. C. E., Pfeil, B., Olsen, A., Metzl, N., Kozyr, A., Fassbender, A., Manke, A., Malczyk, J., Akl, J., Alin, S. R., Bellerby, R. G. J., Borges, A., Boutin, J., Brown, P. J., Cai, W.-J., Chavez, F. P., Chen, A., Cosca, C., Feely, R. A., González-Dávila, M., Goyet, C., Hardman-Mountford, N., Heinze, C., Hoppema, M., Hunt, C. W., Hydes, D., Ishii, M., Johannessen, T., Key, R. M., Körtzinger, A., Landschützer, P., Lauvset, S. K., Lefèvre, N., Lenton, A., Lourantou, A., Merlivat, L., Midorikawa, T., Mintrop, L., Miyazaki, C., Murata, A., Nakadate, A., Nakano, Y., Nakaoka, S., Nojiri, Y., Omar, A. M., Padin, X. A., Park, G.-H., Paterson, K., Perez, F. F., Pierrot, D., Poisson, A., Ríos, A. F., Salisbury, J., Santana-Casiano, J. M., Sarma, V. V. S. S., Schlitzer, R., Schneider, B., Schuster, U., Sieger, R., Skjelvan, I., Steinhoff, T., Suzuki, T., Takahashi, T., Tedesco, K., Telszewski, M., Thomas, H., Tilbrook, B., Vandemark, D., Veness, T., Watson, A. J., Weiss, R., Wong, C. S., and



Yoshikawa-Inoue, H.: Surface Ocean CO2 Atlas (SOCAT) gridded data products, Earth Syst. Sci. Data, 5, 145–153, https://doi.org/10.5194/essd-5-145-2013, 2013.

Takahashi, T., Olafsson, J., Goddard, J.G., Chipman, D.W. and Sutherland, S.C.: Seasonal variation of CO2 and nutrients in the high-latitude surface oceans: A comparative study. Global Biogeochemical Cycles, 7(4), pp.843-878, 1993.

Takahashi, T., Takahashi, T. T., & Sutherland, S. C.: An assessment of the role of the
North Atlantic as a CO2 sink. Philosophical Transactions of the Royal Society of London. Series B: Biological Sciences, 348(1324), 143-152, 1995.

Takahashi, T., Feely, R.A., Weiss, R.F., Wanninkhof, R.H., Chipman, D.W., Sutherland, S.C. and Takahashi, T.T.: Global air-sea flux of CO2: An estimate based on
measurements of sea–air pCO2 difference. Proceedings of the National Academy of Sciences, 94(16), pp.8292-8299, 1997.

Takahashi, T., Sutherland, S.C., Sweeney, C., Poisson, A., Metzl, N., Tilbrook, B., Bates, N., Wanninkhof, R., Feely, R.A., Sabine, C. and Olafsson, J.: Global sea–air
CO2 flux based on climatological surface ocean pCO2, and seasonal biological and temperature effects. Deep Sea Research Part II: Topical Studies in Oceanography, 49(9-10), pp.1601-1622, 2002.

Takahashi, T.: "The fate of industrial carbon dioxide." Science 305, no. 5682: 352-353
1025    2004.

Takahashi, T., Sutherland, S.C., Wanninkhof, R., Sweeney, C., Feely, R.A., Chipman, D.W., Hales, B., Friederich, G., Chavez, F., Sabine, C. and Watson, A.: Climatological mean and decadal change in surface ocean pCO2, and net sea–air CO2 flux over the
global oceans. Deep Sea Research Part II: Topical Studies in Oceanography, 56(8-10), pp.554-577, http://dx.doi.org/10.1016/j.dsr2.2008.12.009. 2009.

Takahashi, T., S. C. Sutherland, D. W. Chipman, J. G. Goddard, C. Ho, T. Newberger, C. Sweeney, and D. R. Munro: Climatological distributions of pH, pCO2, total CO2,
alkalinity, and CaCO3 saturation in the global surface ocean, and temporal changes at selected locations, Mar. Chem., 164, 95–125, doi:10.1016/j.marchem.2014.06.004, 2014.

Takahashi, T.; Sutherland, S. C.; Kozyr, A.: Global Ocean Surface Water Partial
Pressure of CO2 Database: Measurements Performed During 1957-2019 (LDEO Database Version 2019) (NCEI Accession 0160492). Version 9.9. NOAA National



Centers for Environmental Information. Dataset.
https://doi.org/10.3334/CDIAC/OTG.NDP088(V2015) Accessed March 15, 2021.

Tanhua, T., Orr, J. C., Lorenzoni L., and Hansson L.: WMO Bulletin nº : Vol 64 (1) -
2015 (available at https://public.wmo.int/en/resources/bulletin/monitoring-ocean-carbon-
and-ocean-acidification-0), 2015.

Tans, P.P., Berry, J.A., Keeling, R.F.: Oceanic 13C/12C observations: a new window on
ocean CO2 uptake. Global Biogeochemical Cycles 7.2: 353-368, 1993.

Tjiputra, J.F., Olsen, A., Bopp, L., Lenton, A., Pfeil, B., Roy, T., Segschneider, J.,
Totterdell, I., Heinze, C.: Long-term surface pCO2 trends from observations and
models, Tellus B: Chemical and Physical Meteorology, 66:1, 23083, DOI:
10.3402/tellusb.v66.23083, 2014.

Wanninkhof, R., Park, G.-H., Takahashi, T., Sweeney, C., Feely, R., Nojiri, Y., Gruber,
N., Doney, S. C., McKinley, G. A., Lenton, A., Le Quéré, C., Heinze, C., Schwinger, J.,
Graven, H., and Khatiwala, S.: Global ocean carbon uptake: magnitude, variability and
trends, Biogeosciences, 10, 1983–2000, https://doi.org/10.5194/bg-10-1983-2013,
2013.

Weiss, R.: Carbon dioxide in water and seawater: the solubility of non-ideal gas, Mar.
Chem. 2, 203–215, https://doi.org/10.1016/0304-4203(74)90015-2, 1974.