# Peer review of "Updated climatological mean delta fCO2 and net sea—air CO2 flux over the global open ocean regions"

_Earth System Science Data, 2023_

## Author Comment (AC1)

**Response to Reviewer 2**

Title: Updated climatological mean delta fCO2 and net sea–air CO2 flux over the global open ocean regions, by Amanda R. Fay et al.

General comment:

Amanda Fay and CO2-authors revisit the Takahashi et al (2009) climatology using SOCAT version 2022. As there are much more fCO2 data available in SOCAT this is a very good idea to reconstruct such climatology extensively used to constraint atmospheric inversions (a-priori fluxes), methods that reconstruct ocean pCO2 fields (e.g. Rodenbeck et al, 2015) or to validate ocean carbon models (e.g. RECCAP 1 and RECCAP 2 stories). Compared to previous climatology (T-2009), authors found significant differences in the high latitudes (in winter in the SO; in summer in the North Atlantic) and in the south-east Pacific where data were sparse and are still missing in large regions (including in SOCAT-v2024 in progress). The paper is well structured, figures adapted. What is missing is a table (somehow like presented by Takahashi et al 2009, see their table 6) where authors would list: Regions (or biomes), their surface area, the mean DfCO2 and the Flux.

We thank the reviewer for these comments and we have added a table (Table 1) as suggested to provide an overview to the reader of the regional mean values for this updated climatology.

| Biome | $\Delta fCO2$ ($\mu atm$) | Flux (PgC yr$^{-1}$) | Area ($10^6$ km$^2$) |
|---|---|---|---|
| NP Ice | -24.6 (9) | -0.02 (0.02) | 4.2 |
| NP SPSS | -11.5 (12) | -0.11 (0.11) | 12.8 |
| NP Subtropics | -8.2 (16) | -0.40 (0.53) | 47.9 |
| Pacific Equ | 35 (2) | 0.35 (0.03) | 26.4 |
| SP Subtropics | -2.3 (10) | -0.14 (0.31) | 52.7 |
| NA Ice | -19.3 (4) | -0.04 (0.01) | 4.5 |
| NA SPSS | -36.2 (17) | -0.27 (0.08) | 9.7 |
| NA Subtropics | -10.0 (16) | -0.24 (0.26) | 23.4 |
| Atlantic Equ | 14.7 (3) | 0.04 (0.01) | 7.4 |
| SA Subtropics | 5.6 (14) | 0.01 (0.12) | 18.1 |
| Indian Subtropics | -4.5 (8) | -0.18 (0.16) | 35.9 |
| SO STSS | -22.1 (3) | -0.59 (0.06) | 29.6 |
| SO SPSS | -6.0 (6) | -0.21 (0.22) | 30.7 |
| SO Ice | -6.8 (14) | -0.08 (0.12) | 18.7 |

Table 1 Mean annual $\Delta fCO2$ and flux in global open ocean biomes (Fay & McKinley 2014). Value in parentheses is one standard deviation over the 12-month climatology. Area of each biome is also included. NP: North Pacific; SP: South Pacific; NA: North Atlantic; SA: South

Atlantic; SO: Southern Ocean. SPSS: Subpolar seasonally stratified; STSS: Subtropical seasonally stratified. Northern hemisphere subtropical regions are reported to match the regions shown in Figure X (combining the STPS and STSS biomes from Fay & McKinley 2014 into one).

The paper is pleasant to read and suitable for publication in ESSD after few corrections. Below are listed specific comments.

Specific comments:

C-01: Line 22: In Key point: add error/uncertainty on the flux -1.79 PgC yr-1

Thank you for this suggestion. The uncertainty is now included: "-1.79 +/- 0.7 PgC yr$^{-1}$"

C-02: Line 35: In Abstract: add error/uncertainty on the flux -1.79 PgC yr-1

Thank you for this suggestion. The uncertainty is now included: "-1.79 +/- 0.7 PgC yr$^{-1}$"

C-03: Line 41: "now exceed 415 ppm": As atmospheric CO2 increases rapidly, maybe specify the year for the value 415 ppm. Thank you for this suggestion. We have revised the sentence to indicate the year.

C-04: Line 55: Here maybe specify you list the approaches based on observations (i.e. not models):

"multiple approaches based on atmospheric and/or oceanic observations".

We thank the reviewer for this suggestion. We have revised the text to read: "Over the last several decades, multiple approaches based on atmospheric and oceanic observations have been developed to measure the impact of the ocean on the global.."

C-05: Line 83: For Takahashi et al (2009), add also reference to Takahashi et al 2009b (corrigendum). Thank you for pointing out this omission. We have added the requested reference.

C-06: Line 83: I guess you can also refer to Tans et al (1990) who present the first global seasonal map of DpCO2 with the data they had in hand at that time. I think this was the first study that use a "DpCO2 climatology" to constraint a global carbon budget and clearly motivated the next steps (toward a full year ocean pCO2 and flux climatology, Takahashi et al 1997, 2002, 2009...). This also motivated the start of SOCAT first discussed in 2007 (Metzl et al, 2009) and released in 2011.

Thank you for this suggestion and the additional references and context to the history of this field of research and it's motivation. We have added the reference.

C-07: Line 196: You used fCO2, SST, SSS and sea level pressure from SOCAT; for SSS when there is no measurement for a cruise you can recall that you used the SSS from WOA also in SOCAT data files (see Pfeil et al, 2013). Should you indicate somewhere that you select only the Cruises with flags A-B-C-D and data for fCO2 with WOCE flag 2 ?

We have added this information to the SOCAT database discussion (Section 2.1), indicating that we use only SOCAT data with flags A-D and WOCE flag of 2. Added text: "For this analysis we select SOCAT data from cruises with flags A-D and observations with a World Ocean Circulation Experiment (WOCE) flag of 2 (Pfeil et al. 2013)."

C-08: Line 275: add error/uncertainty on the flux for both the normalization method (-1.85 PgC yr-1) and the current method (-1.79 PgC yr-1). We thank the reviewer for this suggestion however we have not amended the text here. The focus of this comparison is simply the impact of the differing normalization on the resulting $\Delta fCO_2$ climatology. Unfortunately, it's difficult to really gauge the meaning of differences of, for example, 0.5µatm when in terms of $\Delta fCO2$, and that is why we here quote the comparison in terms of flux. We hold off on discussing uncertainty until later in the manuscript. If we were to choose an uncertainty value for each of these global flux numbers stated here, they would be the same ($0.7 PgC\ yr^{-1}$ as discussed in Section 4.2). It could even be suggested that this comparison here could provide some clarity to the uncertainty calculation- specifically the portion that is attributed to the normalization method (0.5 $PgC\ yr^{-1}$). The comparison here suggests a much smaller uncertainty due to the correction of available observations to one climatological year than those proposed in previous work (T-2009, Wanninkhof et al. 2013).

C-09: Line 358: Low values of $\Delta fCO2$ in the North Atlantic region also driven by biological activity in Spring-Summer ? (as you mentioned line 445)

We thank the reviewer for this comment. As they point out, we do discuss this spring bloom impact later in the manuscript, in the regional section. Here we are focusing on the annual mean global climatology- looking at broad patterns that emerge on the annual time scale. For that reason, we have opted to keep the text as is here.

C-10: Line 361: Is the near-global mean climatology seasonal curve (figure 2a) useful to discuss ? Would it be better to separate NH and SH (two curves) ?

Thank you for this suggestion. We acknowledge that the mean global curve is indeed not that interesting at first glance, but allowed an interesting look at the bimodal nature and the mean dfCO2 value. We have experimented with your suggestion of a North and South Hemisphere curve and have updated Figure 2 in the manuscript (recreated here).

[Figure]

(a) Global ΔfCO₂ climatology

(b) Annual mean ΔfCO₂ climatology

C-11: Line 552: Chen et al: correct: Chen and Tsunogai (1998). For the large fluxes in the Indian sector and Arabian sea maybe refer to Sabine et al (2000) and/or Sarma et al (2023) ? Thank you for this suggestion. We have added a reference to Sabine et al. who also discuss the upwelling in this region and high flux values.

C-12: Line 560-564: The carbon source in the Southern Ocean during austral winter (Atlantic and Indian sector south of 45S) is likely linked to deep mixing and/or upwelling (import of high DIC in surface layers). We thank the reviewer for this comment and have added additional text to the paragraph: "Another possibility is that the austral winter carbon source is linked to deep mixing and/or upwelling water which would bring an import of high DIC to the surface layers."

C-13: Line 567: "Uncertainties are higher in the Southern Ocean region due to the limited number of observations, particularly in winter". Could you specify "uncertainties": fCO2 reconstruction, fluxes, both ? Compared to T-2009, is there are more data in winter in SOCAT in the SO that would reduce these uncertainties ? We thank the reviewer for this comment. We have opted to rephrase this sentence because it is not appropriate to say "uncertainties are higher" if we aren't calculating regional uncertainties. Our point with this statement was simply that because there are fewer observations in this dynamic area, it requires more interpolation from the method in order to create the monthly climatological map. Thus, we would say we have less confidence in this region due to the known lack of available observations, especially in certain seasons.

To answer your specific question about comparing seasonal data availability to the database used in T-2009, there is indeed much more winter data in SOCAT than there was in the LDEO database that fed into the previous version of this climatology (see figure below). Another view of this can be seen by comparing Figure 1 showing SOCAT to the the LDEOv2019 database (supplementary Figure 1b)- you can see few areas of

the Southern Ocean (outside of the drake passage) that have more than 2 different months sampled for the entire time period in the LDEO version.

[Figure]

Map showing number of additional months per gridcell for data in SOCATv2022 as compared to the LDEO dataset that was used in T-2009. Specifically, that dataset ended with year 2007 and SOCATv2022 includes observations through 2021.

C-14: Line 611: For the comparison of data used in T-2009 and this study, could you show a map where data are in SOCAT-v2022 but not in T-2009. The supp mat (figure S1) shows the LDEOv2019, but might be useful to show the original LDEO when comparing fluxes with T-2009. I guess this is from Takahashi, Sutherland, and Kozyr (2009)

We thank the reviewer for this suggestion. We can understand how a map directly comparing the difference in available observations would help with interpretation of Figure 7 and Supplementary Figure 8b. Here I have plotted maps showing the difference in the number of months with observations, by gridcell, between the two datasets for two seasons: DJF and JJA. The map above shows the difference for all 12 months of the year as a comparison. We have opted to add these figures in the Supplementary rather than the main text.

[Figure]

n.

[Figure]

Map showing number of additional months per gridcell for data in SOCATv2022 as compared to the LDEO dataset that was used in T-2009 for 2 seasons: December, January, and February (DJF) and June, July, and August (JJA). LDEOv2009 dataset includes observations through 2007 and SOCATv2022 includes observations through 2021.

C-15: Line 741: "less data in socat since 2017".  Maybe refer to the Bakker et al (2023)  ? We thank the reviewer for this comment and suggestion. We were aware of the figure showing this trend but had not seen this citation. We have included the reference.

;;;;;;;;;;;; In Figures:

C-16: Figure 7: maybe change the range -50 to 50 µatm to better highlight the differences ? We thank the reviewer for this comment and have considered this option in previous versions of the figure. We elected to use this colorbar range as it matches up with a similar figure in T-2009 (their Figure 11) where they show the differences between that version and the previous version of the climatology. In order to make comparisons with regard to how much change occurred with this update, we opted to maintain a similar colorbar range.

C-17: Figure 8: maybe indicate in the caption that white area in the SO in JJA is because of  ICE extend ? We thank the reviewer for this comment and have added a statement in the caption of this figure.

;;;;;;;;;;;; In references:

C-18: Line 771: Antonov et al : not cited in the MS We thank the reviewer for catching this extra citation. We have removed it from the reference list.

C-19: Line 834: De Vries et al (2023) now published

We have updated the reference.

C-20: Line 1026: Takahashi et al 2009 (check list of authors)

We thank the reviewer for catching this error. We have updated the reference.

;;;;;;;;;;;; In Supplementary Material:

C-21: Table 1: add unit for number listed in this table (µatm/yr)

We thank the reviewer for this comment. We have updated the Table caption.

;;;;;;;;;; Reference added in this review not listed in the paper

Bakker, D., R. Sanders, A. Collins, M. DeGrandpre, T. Gkritzalis, S. Ibánhez, S. Jones, S. Lauvset, N. Metzl, K. O'Brien,  A. Olsen, U. Schuster, T. Steinhoff, M. Telszewski, B. Tilbrook,  D. Wallace, 2023. Case for SOCAT as an integral part of the value chain advising UNFCCC on ocean CO2 uptake http://www.ioccp.org/images/Gnews/2023_A_Case_for_SOCAT.pdf

Metzl, N., Tilbrook, B., Doney, Scott C., Le Quere C., Feely, R. A., Bakker, D.C., Roy S., 2009. Dedication to Dr Taro Takahashi. Deep Sea Res., II, 56, 8-10. https://doi.org/10.1016/j.dsr2.2008.12.039

Sabine, C. L., Wanninkhof, R., Key, R. M., Goyet, C., & Millero, F. J. (2000). Seasonal CO2 fluxes in the tropical and subtropical Indian Ocean. Marine Chemistry, 72(1), 33–53. https://doi.org/10.1016/s0304-4203(00)00064-5

Sarma, V. V. S. S., Sridevi, B., Metzl, N., Patra, P. K., Lachkar, Z., Chakraborty, K., et al. (2023). Air-sea fluxes of CO2 in the Indian Ocean between 1985 and 2018: A synthesis based on observation-based surface CO2, hindcast and atmospheric inversion models. Global Biogeochemical Cycles, 37, e2023GB007694. https://doi.org/10.1029/2023GB007694

Takahashi, T., S.C. Sutherland, and A. Kozyr. 2009. Global Ocean Surface Water Partial Pressure of CO2 Database: Measurements Performed During 1968–2008 (Version 2008). ORNL/CDIAC-152, NDP-088r. Carbon Dioxide Information Analysis Center, Oak Ridge National Laboratory, U.S. Department of Energy, Oak Ridge, Tennessee, doi: 10.3334/CDIAC/otg.ndp088r.

Takahashi, T., S C. Sutherland, R.Wanninkhof, C. Sweeney, R.A. Feely, D. Chipman, B. Hales, G. Friederich, F. Chavez, A. Watson, D. Bakker, U. Schuster,  N.Metzl, H.Y. Inoue, M. Ishii, T. Midorikawa, C.Sabine, M. Hoppema, J.Olafsson, T. Amarson, B.Tilbrook, T. Johannessen, A. Olsen, R. Bellerby, Y. Nojiri, C.S. Wong, B. Delille, N. Bates and H. De Baar, 2009. Corrigendum to Climatological Mean and Decadal Change in Surface Ocean pCO2, and Net Sea-air CO2 Flux over the Global Oceans (DSRII). Deep-Sea Res I, doi:10.1016/j.dsr.2009.07.007

Tans et al.  Observational Contrains on the Global Atmospheric Co2 Budget.Science247,1431-1438(1990).DOI:10.1126/science.247.4949.1431

---

## Author Comment (AC2)

**Response to Reviewer 1**

This manuscript presents an updated version of the Takahashi surface ocean CO2 climatology. Both this and the associated fluxes that are calculated are very useful for a large scientific community. The update is very welcome. I have a few relatively minor comments below.

Specific comments:

Please provide proper uncertainty assessments for the fluxes. You describe in section 4.2 what sources of uncertainty you include, but you appear to have taken numbers from Wanninhof et al (2013) rather than calculating your own numbers based on the %-uncertainty contributed by each term (from their Table 1). A map showing the uncertainties spatially would be highly useful.

We thank the reviewer for these comments. A complete decomposition of uncertainty analysis, including the mapping of uncertainty, is outside the scope of this paper, which focuses on the method of and release of a $\Delta fCO2$ climatology. We have updated the flux discussion with uncertainty estimates drawn from recent publications and direct the reader to pertinent references which focus more on uncertainty analysis. For our uncertainties associated with spatial and temporal extrapolation of $\Delta fCO_2$, we take estimates directly from T-2009. The updated manuscript now uses 13% uncertainty for error associated with $\Delta fCO_2$ and a value of 0.5 PgC yr$^{-1}$ to account for the uncertainty associated with the time normalization step required for a climatology. Updated analysis presented in Wanninkhof 2014 estimates the uncertainty on gas transfer velocity to be 20%. Finally, Flux analysis presented in Fay & Gregor et al. 2021 estimate uncertainty on the wind reanalysis product to be 0.09 PgC yr$^{-1}$. Lastly, we maintain the river carbon uncertainty of 0.2 PgC yr$^{-1}$ as presented in Wanninkhof et al. 2013 (Jacobson et al. 2007). These estimates, summed in quadrature, result in a total uncertainty estimate of 0.7 PgC yr$^{-1}$. This value has been added in the main text and in the Key Points.

Throughout the manuscript several averages are presented. Please provide also a standard deviation for all of them.

We thank the reviewer for this comment. We have added uncertainty (1 standard deviation) estimates to key averages when they are first presented in the text, but left them out in subsequent mention as we feel that it would bog down the reader by adding these values throughout. Specifically, for Section 4 where we discuss regional means, often in the same sentence the amplitude of the seasonal cycle is presented which is an alternative way to visualize the standard deviation of an annual mean value (i.e. a large amplitude seasonal cycle would ultimately have a larger standard deviation for the annual mean). Rather than including these values in the text, we have added a table of mean $\Delta fCO2$ and flux values by biome as suggested by Reviewer 2 (Table 1) and we have included the standard deviation over the 12 months there.

If the reviewer is specifically interested in seeing the standard deviation spatially for the biomes scale means, we again think that including these specific numbers would bog down a reader. Instead, by showing the maps, a reader can quickly assess how variable the values are within a biome.

Overall, we strongly agree that showing uncertainties is important in our field, and we have included an expanded discussion of the uncertainty on our flux estimate, however producing specific standard deviation values for every stated mean within this manuscript is not the ideal way to communicate uncertainty.

The SOCAT data product was first released in 2011. Thank you for this correction. We have changed the year in the manuscript.

It is very difficult to see the lowest numbers on Figure 1a. Could you make 0 white as in Figure 1b? We thank the reviewer for this comment. The areas with 0 observations are now white, as in Figure 1b. This has improved the figure to make the low values more visible.

Did you test your assumption/hypothesis that there is no trend in \Delta_fCO_2 over the 40 year period? A figure in the supplement showing this would be nice I think.

We thank the reviewer for this question. We did quite an extensive exploration regarding this choice of time normalization, much of which we included in the methods discussion section of the paper. We experimented with using the exact same method as T-2009 which was a trend of 1.5µatm/yr globally as well as a time-varying normalization trend (for example, 1.5µatm/yr prior to 2000 and then 2.0µatm/yr after 2000). None of these versions of the climatology resulted in significantly different global means or seasonal cycle patterns for fCO2 or flux. There were regional differences, but typically that was most prominent in regions with severely limited data (for example the Indian subtropics or the high latitude ice regions). We acknowledge that globally, there is indeed an increasing trend in $dfCO_2$, and therefore a growing ocean carbon sink. However, given the limitations of the available data, no choice is perfect for creating a climatology. We have opted for this method as it is a straightforward method given the available observations in SOCAT (i.e. co-located ocean and atmosphere fCO2 values). Below are time series of all available $dfCO_2$ values that go into our climatology (from SOCATv2022), broken down by biome. Recorded on top of each subplot is a trend in the $dfCO_2$ values from 1980-2021. Globally the trend is -0.07 µatm/yr. The biomes trends vary from -0.09 µatm/yr (in the NP SPSS biome) to 0.18 µatm/yr (in the NP ICE where there is very limited available data before year 2000). None of these trend values are statistically different from zero given 1 sigma confidence intervals. Therefore, we have strong confidence that our assumption is appropriate for the project goal and the available data.

[Figure]

How big is the area of the Arctic Ocean you make land?

It is not so much that we "assign land" to the arctic region as much as we do not produce an estimated ΔfCO2 climatological value for these high latitude regions. This is predominantly due to the lack of available observations to base an estimate on, thus making it very likely a highly uncertain estimate. As stated in the manuscript, just under 10% of the global ocean does not have a value reported in this climatology. Considering ocean areas north of 50N latitude, there are $9.78 \times 10^6$ square km that do not have a reported climatological value in our product.

Line 490: It is unclear which quantity you are referring to here.

We thank the reviewer for this comment. The quantity we are referring to here is the estimated flux for the ocean areas which do not have a reported value in our near-global

climatology. Using the 12-month climatological coastal and high latitude product (Landschützer et al. 2020), we are able to calculate the flux (ocean uptake of carbon) that we are missing by not accounting for these areas. That number is not the same for each month, but varies over the 12 months of the climatology. We have adjusted the text of this sentence to hopefully clarify the meaning. It now reads, "This flux adjustment for missing areas of this climatology varies throughout the seasonal cycle, ranging from -0.43 to -0.31 PgC yr$^{-1}$ during the 12 months of the climatology."

Line 732-735: This needs revision for clarity. I assumed that the Southern and Northern hemispheres would add up to the global, but it does not (since the tropical area is missing). We thank the reviewer for catching this. In fact, the 3 stated numbers would add up to -1.88 PgC (-1.19+ -1.04 +0.35) which is just slightly more uptake than the mean value we reported in the manuscript. The reason for this is that there are a few areas that do not fall in the defined biomes, but do have an estimated flux value in this climatology, and those regions make up 0.08 PgC/yr of flux. These areas include regions of the Gulf of Mexico as well as areas of the Arabian Sea. To make this conclusion section more clear, we edited the sentences and just split the ocean at +/-30 latitude and quoted the fluxes for those regions. It still communicates the same message- that the southern hemisphere ocean regions are the largest uptake region and the equatorial/tropical region is an area of efflux of carbon. The edited sentence now reads, "Of the major ocean basins, the Southern Hemisphere ocean (south of 30S) is the largest $CO_2$ sink, taking up 1.22 PgC yr$^{-1}$, while the Northern Hemisphere ocean (north of 30N) takes up 0.93 PgC yr$^{-1}$. The equatorial ocean region acts as the only year-round region of carbon efflux to the atmosphere and emits 0.36 PgC to the atmosphere annually."

I think including the section "LDEO flux" from the supplement in the main text would make sense.

We thank the reviewer for this suggestion. We have experimented with the structuring of this paper in many previous versions and after much deliberation and comments from internal reviewers we have decided to leave all discussion of the results from the LDEO database in the supplementary section in order to reduce confusion with the results discussed here, which are the primary findings and official updated climatology. The information included in the supplementary, discussing the results using the LDEO database, is a tribute to Taro's impressive work and legacy creating, maintaining, and utilizing that database. Since it is no longer supported or updated, we opted to not include it in the main paper.

I often find the manuscript a bit difficult to read. There are many very long and cumbersome sentences that I struggle to understand. There is also a rather excessive use of semicolons. I know many like semicolons, and I admit to having a particularly strong dislike of them, but they do not aid reading. A semicolon, most often, replaces a word (the word you would need if you used a comma instead). As a non-native reader my brain keeps stopping and trying to identify what the word is. I can't seem to fully grasp what the sentence says without mentally putting that word into it. This makes for slow and frustrating reading. I would therefore urge you to go through the text and

simplify the language and ensure better readability. Many times shorter sentences would do the trick.

We thank the reviewer for this comment and suggestion. We have edited the manuscript throughout to shorten sentences and limit semicolon use. We hope this has improved the readability for all readers.